# Spotlight on G-Quadruplexes: From Structure and Modulation to Physiological and Pathological Roles

**DOI:** 10.3390/ijms25063162

**Published:** 2024-03-09

**Authors:** Maria Chiara Dell’Oca, Roberto Quadri, Giulia Maria Bernini, Luca Menin, Lavinia Grasso, Diego Rondelli, Ozge Yazici, Sarah Sertic, Federica Marini, Achille Pellicioli, Marco Muzi-Falconi, Federico Lazzaro

**Affiliations:** Department of Biosciences, University of Milan, 20133 Milan, Italy; mariachiara.delloca@studenti.unimi.it (M.C.D.); giulia.bernini@unimi.it (G.M.B.); luca.menin@unimi.it (L.M.); lavinia.grasso@unimi.it (L.G.); diego.rondelli@unimi.it (D.R.); ozge.yazici@unimi.it (O.Y.); sarah.sertic@unimi.it (S.S.); federica.marini@unimi.it (F.M.); achille.pellicioli@unimi.it (A.P.); marco.muzifalconi@unimi.it (M.M.-F.)

**Keywords:** G4, nucleic acids structures, DNA, RNA

## Abstract

G-quadruplexes or G4s are non-canonical secondary structures of nucleic acids characterized by guanines arranged in stacked tetraplex arrays. Decades of research into these peculiar assemblies of DNA and RNA, fueled by the development and optimization of a vast array of techniques and assays, has resulted in a large amount of information regarding their structure, stability, localization, and biological significance in native systems. A plethora of articles have reported the roles of G-quadruplexes in multiple pathways across several species, ranging from gene expression regulation to RNA biogenesis and trafficking, DNA replication, and genome maintenance. Crucially, a large amount of experimental evidence has highlighted the roles of G-quadruplexes in cancer biology and other pathologies, pointing at these structurally unique guanine assemblies as amenable drug targets. Given the rapid expansion of this field of research, this review aims at summarizing all the relevant aspects of G-quadruplex biology by combining and discussing results from seminal works as well as more recent and cutting-edge experimental evidence. Additionally, the most common methodologies used to study G4s are presented to aid the reader in critically interpreting and integrating experimental data.

## 1. Introduction

Nucleic acids are the biopolymers responsible for storage of genetic information, its heritability, and decoding to effectively build the components of an organism. While most of their functions have been linked to their canonical secondary structures, mounting experimental evidence has revealed the existence and functional relevance of a wide variety of alternative conformations of DNA and RNA, among which the G-quadruplex has been the most extensively studied. The following paragraphs expand upon seminal reviews in the field [1,2,3] by providing a summary of relevant and more recent literature pertaining to the structure, stability, dynamics, and physiological and pathological relevance of this non-canonical secondary structure.

## 2. Overview of the G-Quadruplex

### Discovery and Structure of the G-Quadruplex

Research on G-quadruplexes and their structural unit, the G-tetrad or G-quartet, is a half-a-century old endeavor (Figure 1). The first experimental evidence of the self-assembly capabilities of guanines dates back to the 1910s, when it was reported that guanylic acid solutions at high concentrations formed a gel [4,5,6].

Five decades later, the mechanism behind this curious phenomenon was revealed in an X-ray crystallography experiment, where the guanines of 5′-GMP were found to associate into a helical higher-order structure, presumed to be formed by the stacking of planar structures of guanine quartets held in place by hydrogen bonds [7]. This structural hypothesis of the G-tetrad was later built upon using X-ray diffraction data of polyinosinic acid [8,9], which is structurally similar to poly(G). In the late 1980s, new studies using oligonucleotides corresponding to the guanine-rich immunoglobulin switch regions and telomeric sequences postulated that these formed four-stranded assemblies compatible with stacked G-tetrads [10,11]. In addition to using sequences from real genomic regions notorious for their tandem GC repeats, the authors of these studies provided new compelling hypotheses on the involvement of this poorly characterized secondary structure in key biological processes, such as meiotic chromosome pairing before crossing over [10] and telomere regulation [11]. Since then, new insights into the structure of G-quadruplexes have been made thanks to a wide variety of experimental techniques (reviewed in ref. [12]), while the function of this assembly has been probed in biological systems. Indeed, shortly after in a seminal work it was found that oligonucleotides, bearing ciliate *Oxytricha nova* telomeric repeats folded into G-quadruplexes, inhibited telomerase activity in vitro [13], thus linking this structure to cancer biology and anticancer therapy. Nowadays, a consensus for the intramolecular G-quadruplex (also known as G4 or tetraplex) has been reached (Figure 2a), and its structure is understood to amount to at least two stacked planar G-tetrads, each consisting of four guanines interacting with one another forming non-canonical Hoogsteen bonds. More specifically, within the plane of a G-tetrad, four guanines are placed according to a four-fold rotation axis, exposing to one another compatible pairs of chemical groups that function as electron donor and acceptor. In this configuration, the electron-poor hydrogens of the -NH groups of the guanine base establish hydrogen bonds with the electron rich oxygens and nitrogens of the nearby guanine. In this planar array, stacked G-tetrads are further linked by van der Waals forces thanks to the π electrons of the aromatic rings from each base (reviewed in ref. [14,15]). The whole structure is additionally stabilized by metal ions, particularly monovalent alkali cations such as Na^+^ and K^+^, which are positioned at the central axis and coordinate the electron-rich oxygen atoms of the nearby guanilyl bases [16] (Figure 2b,d). Several in vitro studies have reported the ability of both monovalent and divalent cations to aid G4 folding and stability, albeit to different extents (reviewed in ref. [17]). The presence of four bases per G-tetrad implies that the G-quadruplex is a four-stranded structure, with each strand either being part of the same nucleic acid molecule (intramolecular G4s) or belonging to different molecules (intermolecular G4s). Additionally, the phosphodiester backbone, linking guanilyl bases of consecutive G-tetrads, has intrinsic directionality, meaning that a strand can be denoted as having a parallel or antiparallel orientation to each of the neighboring ones (Figure 2c). Thus, G-quadruplexes can be referred to as parallel, antiparallel, or hybrid, the latter corresponding to only one G-strand running in the opposite direction with respect to the other three.

A notorious example of a parallel G-quadruplex structure comes from the determination of the human telomeric repeat assembly in the presence of potassium [18]. Noteworthily, in the case of intramolecular G4s, two strands participating in a G-quadruplex must be linked by a loop. Numerous loop connections have been described, with the simplest ones being the propeller, lateral, and diagonal loops (reviewed in ref. [14,19]; Figure 2c). Interestingly, it has been demonstrated that loop length and sequence affect the stability and folding of the tetraplex structure [20,21,22,23,24,25]. An additional element to be considered when describing the structure of a G-quadruplex is the type of grooves of the assembly, which are defined by the *anti* or *syn* conformations of the ribonucleotides or deoxyribonucleotides linked to the guanines participating in the G-tetrad. These conformations refer to the torsion angles formed by the glycosidic bond between the sugar and the nucleobase moieties. Depending on the configurations of the nucleotides of adjacent guanines in a tetrad, grooves are defined as wide, medium, or narrow, with the first and last of these corresponding to arrangements where aligned nucleotides have opposite glycosidic bond angles [15,26,27,28]. Although all the topologically possible combinations of N-glycosidic bond conformations and canonical loops have been determined [26], only a subset of these have been experimentally verified [29,30,31,32]. To complicate things even further, a number of non-canonical G-quadruplex assemblies have been observed over the years (reviewed in ref. [33]). While most G-quadruplexes have been reported to have a right-handed helical twist, left-handed G4s [34] and even hybrid G4s [35] have also been observed. Left-handed G-quadruplexes can be formed by at least two distinct minimal sequence motives of 12 nt [36,37]. Interestingly, the GTGGTGGTGGTG motif, which is highly abundant in the human genome, was not only shown to independently form left-handed G4 structures, but also to drive the formation of left-handed conformations from several other sequences, when attached to them [36]. Machine learning methods to classify right- and left-handed G4s based on torsional angles have been recently explored [38]. Moreover, bulges or protrusions of nucleotides from the G-tract [39,40], vacant guanine spots inside G-tetrads, snapback loops that fill in the empty spot [41,42,43,44,45,46], and D-shaped loops connecting guanines that are not contiguous in sequence but that are part of the same G-tract [47,48], are only some examples of our current understanding of unusual G-quadruplex topologies. All in all, depending on the combination of the number of nucleic acid molecules that participate in the formation of a G-quadruplex, the relative orientation of its backbone strands, and the presence and topology of connecting loops and bulges, it is evident that a wide variety of G-quadruplex conformations is possible. Crucially, it has been determined that the same G-quadruplex-forming sequence can give rise to two or more coexisting G4 structures in solution [30,49,50,51]. Interestingly, the type of cation in a buffer was shown to influence quadruplex conformation in vitro (reviewed in ref. [17]). In addition, different structures show variable affinity towards G-quadruplex-binding proteins [52,53] and small molecules [54,55,56], and this can be the basis for the selection of molecules that are specific for different structures. Given the abundance of polymorphic G-quadruplex structures, it is thus tempting to assert that cells can discriminate between different G4 topologies, which would then be linked to distinct biological roles in live cells [19], although the extent to which this is true is yet to be elucidated.

**Figure 2 ijms-25-03162-f002:**
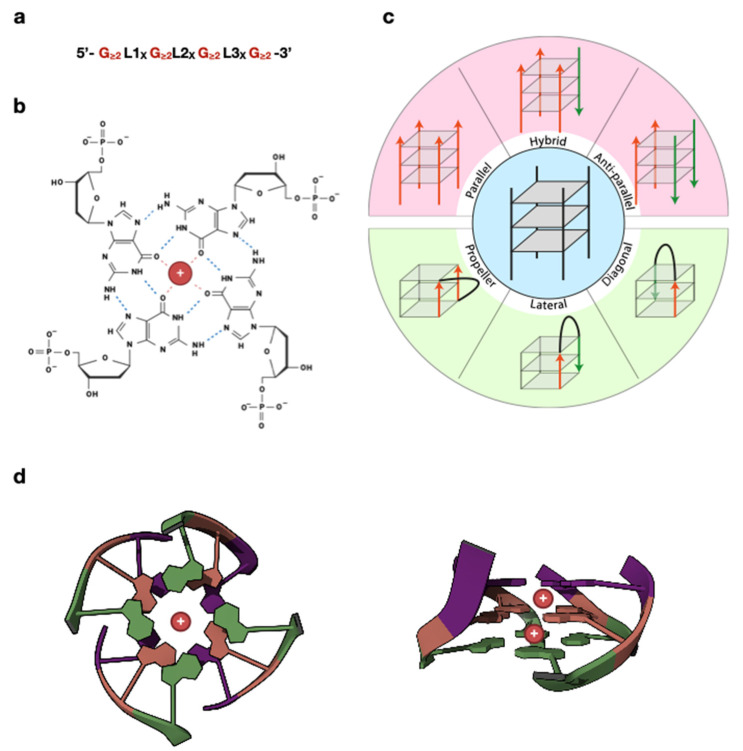
Structural organization of G-quadruplexes. (**a**) Basic sequence of an intramolecular G-quadruplex. (**b**) Structure of a DNA G-quartet, the basic structural element of a canonical G-quadruplex. Four coplanar guanines establish Hoogsteen bonds between their complementary surfaces, while the electron-rich O atoms are stabilized through coordination of the central monovalent cation. Multiple G-quartets are stacked upon one another via π-π orbital interaction of their aromatic systems. (**c**) Schematic representation of possible strand and loop orientations within a G-quadruplex. G4s are categorized as parallel when all four strands of their G-quartets have the same orientations, antiparallel when two pairs of strands run in opposite directions, or hybrid when three out of four strands are codirectional. G-quadruplex strands are connected by loop structures, the most common being the propeller (**bottom left**), the lateral (**center bottom**), and the diagonal loops (**bottom right**). (**d**) Top (**left panel)** and lateral (**right panel**) views of human telomere structure (PDB ID: 2HY9 [57]) modified with Mol* (https://doi.org/10.1093/nar/gkab314). Different colors represent different G-quartets.

Over the years, significant strides have been made in providing structural information of sequences harboring G-quadruplexes. As was expected from their G-rich repeats, structural studies have revealed that human telomeric repeats fold into G-quadruplexes in vitro [18]. Besides telomeres, sequences shown to fold into G-quadruplexes in vitro were found in promoters of key genes, particularly in the nuclease-hypersensitive element III of human *c-myc* [58,59,60] and a regulatory element 87 nucleotides upstream of the transcription start site of human *c-kit* [46,61]. In addition, in human minisatellites CEB1 and CEB25, G4 structures were solved by NMR [62,63]. Despite mounting evidence of G4 structures from G-rich sequences, it is not guaranteed that tetraplexes can fold at corresponding genomic sites inside cells. A key aspect of most in vitro studies is that G-quadruplex structures are determined using single-stranded oligonucleotides. It is important to consider that in chromosomal DNA, the tetraplex structure is in competition with the standard double-helix conformation, with the displaced C-rich strand available to re-establish Watson–Crick complementarity. Indeed, it was found that, in the presence of the complementary strand, some oligonucleotides prone to G-quadruplex formation tended to form duplexes instead [21,64,65,66,67,68,69,70,71,72]. This suggests that G-quadruplexes do not fold automatically in chromosomal DNA unless the double helix is unwound or denatured to expose the guanines for G-tetrad formation. One possibility of that occurring is when DNA is subjected to negative supercoiling; however, it has been demonstrated that this alone cannot induce G-quadruplex folding in plasmids [73]. Contrastingly, single-molecule imaging experiments in cells revealed that G-quadruplexes are indeed formed upon dsDNA unwinding during DNA replication [74]. Furthermore, when the complementary C-rich strand is independently stabilized, G-quadruplex formation is favored. This was demonstrated in vivo with the so-called G-loop, a structure where R-loop formation on the transcribed strand is coupled to G-quadruplex folding on the non-template strand [75]. However, the opposite is true when considering the case of i-motifs. Similarly to guanine stretches, a series of cytidine residues in DNA or RNA can also associate in four-stranded structures called i-motifs. These are secondary nucleic acid structures consisting of four strands stabilized by hemi-protonated and intercalated cytosine base pairs (C:C+) [76,77]. It has recently been widely demonstrated that the complementary strand of any G-quadruplex-forming sequence is prone to forming i-motifs (reviewed in ref. [76,78]). Importantly, the stabilization of G-quadruplexes using small molecules was shown to destabilize the i-motifs, and vice versa. This suggests that these structures are interdependent [79]. Owing to their distinctive physicochemical properties, these i-motif structures have garnered considerable attention as novel targets for drug development (reviewed in ref. [80]); however, a thorough discussion of these secondary structures is beyond the scope of this review. Another factor that ties G4 folding to polymerase activity is the increased stability of the tetraplex in nanocages mimicking the confined space in the exit channel of polymerases [81]. G-quadruplex folding was also found to be favored in molecular crowding conditions [82,83,84], likely more representative of the in vivo environment of chromosomal DNA. Another aspect to consider is the local chromatin accessibility, a feature tightly regulated by chromatin factors and nucleosome positioning. Indeed, it was found that G-quadruplexes in native chromatin of human cells often colocalized with markers of euchromatin (H3K4me3 and RNApol II), which are mostly nucleosome-depleted [85], and that G4 formation likely displaces nucleosomes in vitro [86]. While the concept of non-B DNA structures being excluded by nucleosomes and accumulating in open chromatin is compelling, it needs further investigation. Overall, mounting evidence indicates that G4 folding in vivo is possible and facilitated by exposure of ssDNA, stabilization of the complementary C-rich strand, the crowded conditions genomic DNA can find in cells, confined spaces resembling the exit channel of polymerases, and possibly nucleosome-depleted accessible chromatin. As for RNA G-quadruplexes, in vitro studies determined that ribonucleotides could also fold in tetraplex assemblies. Intriguingly, given the single-stranded nature of most RNAs, these structures were predicted to fold more easily than in DNA, as well as being exceptionally stable [87,88,89]. In addition to proving that G-quadruplex folding is feasible in cells, and thus allowing further investigations into the biological function of these structures, most of the abovementioned conditions are compatible with transcription, telomere elongation, and DNA replication. The biological relevance of G-quadruplexes will be discussed in a later section. 

## 3. Tools to Study G-Quadruplexes 

Depending on the biological question, a plethora of protocols, instruments, and probes can be used to determine structural features or biological functions of G-quadruplexes. Understanding the type of data each one produces, as well as advantages and disadvantages, is fundamental when trying to piece evidence together into a comprehensive view of G-quadruplex biology. Knowledge of the most common types of assays can also help in validating results or when developing or improving an experimental protocol. The following sections will provide a brief description of the techniques that have provided hallmark results in the past, as well as those that are widespread in today’s scientific reports pertaining G4 biology (Table 1). Newly developed probes and protocols will also be briefly discussed to provide insight into how the field is being improved.

### 3.1. In Vitro Structural Studies

In vitro studies of G-quadruplex structure and stability remain a fundamental step in probing the morphology and stability of a tetraplex. As mentioned, biochemical and structural investigations were first used to discover the structure of telomeric G-quadruplexes. These types of studies were fundamental to assess the wide variety of canonical and non-canonical G-quadruplex structures, paving the way for the concept of G-quadruplex polymorphism. Furthermore, seminal investigations into the dynamics of G-quadruplex forming oligonucleotides were the basis for understanding the structural elements and environmental conditions (cations, pH, and molecular crowding) that impact on G-quadruplex dynamics. These insights are still valuable in current research, as structural information can be used for rational ligand design and molecular simulations. Among all in vitro assays that have been used to probe G-quadruplex morphology, NMR spectroscopy (reviewed in ref. [90]) and X-ray crystallography (reviewed in ref. [91]) are the most common. The latter allows structural determination of DNA or RNA G-quadruplexes, alone or in complex with proteins or ligands, at Angstrom resolution. Despite the potential of this technique, its major bottleneck is the formation of G-quadruplex-containing crystals suitable for analysis. X-ray crystallography requires that the target molecule is found in a highly ordered crystalline array, which is challenging to obtain. Several conditions need to be probed in G-quadruplex crystallization protocols to obtain a viable crystal [91,92]. On the other hand, the need for crystallization is completely bypassed in NMR spectroscopy, which offers the unique advantages of assaying G-quadruplex structure and dynamics in solution at close to physiological conditions [90,93]. In the NMR experiments, the appearance of an imino peak in the 10.0–12.5 ppm range is indicative of the G-tetrad formation [93] and can be used to detect more than one G-quadruplex conformation at the same time [94], although this may result in extensive spectral overlap. When the objective is to study a single structure from a sequence capable of folding into multiple ones, such a sequence is modified accordingly, for example, by changing the loop length or the sequences flanking the G-quadruplex [90]. Besides the oligo sequence, a desired G-quadruplex morphology can be selected by adjusting buffer and experimental conditions [90] or by removing all undesired structures from the sample using size exclusion chromatography [95]. NMR is also suited to monitoring G-quadruplex stability over time or structural changes upon addition of a ligand, allowing a deeper insight into the kinetics and dynamics of a particular conformation, as well as aiding ligand design [90]. Interestingly, some NMR protocols can also be performed in living cells, as is the case for a ^19^F NMR spectroscopy study on human telomeric RNA G-quadruplexes injected into *X. laevis* oocytes [96]. Crucially, the study where this technique was described also unequivocally reported the existence of folded RNA G-quadruplexes in vivo [96], which had been previously debated [97]. Despite the widespread use of crystallography and NMR structural studies, their applicability is currently beyond the reach of investigating higher-order G-quadruplex structures due to their high topological flexibility, a feat that would provide unvaluable data on G4 positioning into a close-to-native DNA duplex. A recent study [98] provided a possible solution to this problem. By combining small angle X-ray scattering (SAXS), molecular dynamics simulations, previously solved G-quadruplex structures, and a 7.4 Å resolution cryo-EM model, the integrative approach yielded a plausible structural model of a parallel G4 embedded into duplex DNA with a polyd(T) ssDNA stretch. This tertiary DNA structure closely mimics that of promoter G-quadruplexes and suggests that guanine tetraplexes are found stacked coaxially to the dsDNA duplex surface, rather than protruding outside of the double helix as previously thought [98].

An alternative to X-ray crystallography and NMR spectroscopy that does not require expensive equipment is circular dichroism (CD) spectroscopy, which was a pioneering method used to study DNA secondary structures. Although providing much less informative than the two techniques described above, CD spectroscopy can be used to distinguish between parallel and antiparallel tetraplex conformations based on characteristic CD spectra of the two conformations; parallel G-quadruplexes exhibit a sharp ellipticity maximum at 260 nm and a minimum at 240 nm, while antiparallel structures show a maximum and minimum at about 290 nm and 265 nm, respectively. These stark differences in CD spectra are the result of the different N-glycosidic torsion angles of the strand participating in G-quadruplex formation (all *anti* in parallel G-quadruplexes, and alternate *anti-syn* in antiparallel ones), leading to differently stacked G-tetrads [14,99]. CD spectra are easily interpreted using these criteria, especially when studying telomeric G-quadruplexes; however, non-canonical conformations and structures that do not conform to telomeric G-quadruplexes may yield different CD spectra and prevent scientists from assigning strand orientation to the structure. Nevertheless, outside of these cases, CD spectroscopy offers the possibility to calculate the melting temperature, thus providing additional information on the stability of the assembly [14]. On the basis of the characteristic transition of UV-visible absorbance for G-quadruplexes at around 295 nm, three biophysical methods for the characterization of G4 structures in vitro are commonly used: isothermal differential spectrum (IDS), thermal differential spectrum (TDS), and UV melting (reviewed in ref. [100]). In particular, UV melting experiments are performed to assay G-quadruplex thermal melting by measuring UV absorbance of the sample at 295 nm at increasing temperatures [101]. In combination with these techniques, studies using ultraviolet resonance Raman spectroscopy [102] and differential scanning calorimetry [103] can provide important information on the structure and dynamics of G-quadruplexes. 

Alternatively, G-quadruplex unfolding can be monitored with fluorescent probes linked to the extremities of oligonucleotides. For example, Fluorescence Resonance Energy Transfer (FRET) has been adapted for G-quadruplex stability experiments; FRET efficiency (acceptor fluorescence intensity over donor fluorescence emission) can be used to measure the distance of the 3′ and 5′ ends of a ssDNA, which is minimal when a G-quadruplex is present [65,104,105]. Although particular care should be taken in the choice of fluorophores and in minimizing their impact on G-quadruplex foldability, FRET provides the unique advantage of absolute distance measurement as direct evidence of tetraplex folding and unfolding [104] and has evolved to reach single molecule resolution [66,106]. Recently, evolutions of this technique, such as FRET-MC (melting competition) assay and its isothermal version, Iso-FRET, have been proposed to analyze quadruplex formation in vitro [107,108]. Moreover single-molecule force spectroscopy techniques, include optical tweezers (OT), magnetic tweezers (MT), and atomic force microscopy (AFM) allow real-time detection of folding/unfolding dynamics and discovery of different G4 topologies (reviewed in ref. [109]).

### 3.2. Bioinformatic Prediction of G-Quadruplexes and Polymerase Stop Assays

In vitro biophysical studies have provided instrumental information on G-quadruplex structure variability, stability of different tetraplex forms, and how it is impacted by sequence elements. This allowed the definition of rules to predict whether a given sequence can fold into a G-quadruplex, laying the bases of bioinformatic predictions. The earliest works in this direction used a simple sequence model built on the assumption that intramolecular G-quadruplexes require four short stretches of at least three consecutive guanines separated by short spacers that would become the loops of the assembly [110,111]. In other words, the basic sequence defined as having the potential to fold into a G-quadruplex had the following structure: G_3–5_L1_1–7_G_3–5_L2_1–7_G_3–5_L3_1–7_G_3–5_, where L1, L2, and L3 are the three loops connecting the guanines that are assembled into G-tetrads. The maximum length of these connecting sequences was capped at seven nucleotides to reduce complexity [111]. Even with this low complexity model, the Quadparser algorithm indicated that as many as about 375,000 non-overlapping sequences in the human genome could potentially form a G-quadruplex [110,111]. In the following years, this number has risen thanks to improvements in algorithms and putative quadruplex sequences (PQS) models (reviewed in ref. [112]). Despite the widespread adoption of the original strategy based on the abovementioned consensus motif [113,114,115], new methods have since been developed to allow the detection of non-canonical G-quadruplex structures. For example, the presence of a limited number of bulges, mismatches, and long loops in a sequence does not automatically exclude it from G-quadruplex propensity evaluation in pqsfinder, although a score is assigned to each identified sequence to penalize such imperfections [116]. Other detection tools, such as G4Hunter, assign the score using G richness and G skewness [117]. More recent algorithms have incorporated machine learning models [118,119] and/or are also trained on a dataset of sequences confirmed to form G-quadruplexes in vitro [116,117]. Overall, as new tools are developed to more accurately predict putative canonical and non-canonical G4 sequences, it is evident that the abovementioned sequence model used by the Quadparser algorithm has become obsolete.

Additionally, other bioinformatic tools have been used to predict the three-dimensional conformation of G-quadruplexes from a given sequence. Indeed, inter- and intramolecular G-quadruplex structures can be generated in silico using web tools such as 3D-NuS [120]. This modeling could be used to explore the dynamics of different G-quadruplexes and the docking of proteins and ligands.

Furthermore, the efficacy of some PQS prediction algorithms was tested using the output of the high-throughput G4-seq method [121], which consists of a modified Illumina sequencing protocol where polymerase arrest upon encountering a folded G-quadruplex in the template is identified by a drop in sequencing quality scores. Only the dip in Q-scores specific for G-quadruplexes was selected thanks to the comparison between a sequencing run in standard conditions and one where G-quadruplex stabilizing factors (either K^+^ or the small-molecule ligands such as pyridostatin or PhenDC3 were added to the sequencing buffer). G4-seq of the human genome identified more than 700,000 G-quadruplex forming sites, of which about 70% were not predicted by the standard G4 motif-based Quadparser algorithm [110,121], possibly comprising non-canonical G4 structures escaping the algorithm folding rule (G_3_+N_1–7_G3+N_1–7_G3+N_1–7_G3+), like those with loops longer than seven bases, bulges in the G-tracts, or only two G-tetrads. On the contrary, the original G4-seq method failed to detect 27% in PDS and 40% in K+ of Quadparser-predicted canonical G4-forming motifs [121]. Even when considering predicted potential G4-forming motifs with loop lengths up to 12 nucleotides, 37% of them still evaded G4-seq in PDS outputs [122]. This inconsistency may be mainly explained by inadequate sequencing coverage in certain GC-rich genomic regions (due to inefficient amplification at stable G4s in Na+ and PCR biases during library preparation), PDS stabilization performed in Na+, and low resolution of the observed G4 motifs and consequent merging of proximal G4 motifs of the original experiment. Additionally, limited G4 stability, binding specificities of the employed G4 ligands, and the in vitro experimental conditions may account for a fraction of false negatives. To overcome these limitations, improvements have been introduced in the second-generation method that have successfully increased the specificity of the assay in K^+^ PDS. In fact, the refined protocol detected ~95% of human canonical Quadparser G4s and 84% of the ~706 k potential G4-forming sequences with loops as long as 12 nt [122]. 

The assay was also adapted for transcriptomic analysis; in rG4-seq reverse transcriptase stalling is induced when G-quadruplexes are folded in vitro on template mRNA upon addition of K^+^ or K^+^ and pyridostatin (PDS) [123]. After retrotranscription and Illumina sequencing, the sequences that form G-quadruplexes are detected as a drop in coverage when compared to the same assay performed with Li^+^, a known G-quadruplex destabilizing agent [123]. Analysis of the HeLa transcriptome revealed that as much as 88% of detected reverse transcriptase arrests correspond to non-canonical G-quadruplexes [123]. It is evident that both G4-seq and rG4-seq can potentially detect tetraplex structures without prior knowledge of their sequence motif or overall fold, while traditional bioinformatics tools are based on a model that cannot be applied reliably to all possible G-quadruplexes. Despite this, both polymerase-stop assays and bioinformatic predictions of G-quadruplexes offer a limited view of the role of these structures in genomes and transcriptomes, because the presence of these folded structures must be validated in vivo to provide any meaningful insight. In addition, both approaches offer potentially biased results, since most algorithms are built on a sequence model that cannot take into account G-quadruplex structures that are yet to be studied, while G4-seq and rG4-seq offer only indirect proof of G-quadruplex formation and use small-molecule ligands that may have binding preference for certain folds over others. These issues, however, were partially addressed in the original studies; rG4-seq results showed more pronounced reverse transcriptase stalling in the K^+^-PDS condition with respect to the K^+^ only experiment [123], while G4-seq data with PDS was largely overlapping with that of the alternative G4 stabilizer molecule PhenDC3 [121,124].

A further improvement may come in the next years from innovative sequencing technologies such as nanopore sequencing. Indeed, a recent work proved that non-B DNA structures, including G-quadruplexes, can be predicted from whole genome nanopore sequencing data, based on the timing of DNA translocation of non-B compared with B DNA [125]. These findings represent a crucial step towards direct G4 detection in cellulo and in vivo, which could add an additional step in the study of these structures.

### 3.3. Antibody-Based Methods for G4 Detection

While in vitro and in silico studies of G-quadruplexes are fundamental to understand the structure, stability, folding dynamics, and prevalence of G4-forming sequences in nucleic acids, the need for a dissection of their functional relevance has fueled the development of tools to detect these structures in vivo. Within cells, the complex interplay between guanine tetraplexes, native chromatin context, G-quadruplex-binding proteins, and those involved in genome maintenance and RNA biogenesis results in G-quadruplexes modulating a molecular process or pathway to ultimately produce a biologically relevant effect on the system. To link such effects to G-quadruplexes, the most common strategies rely on designing a probe, antibody, or small molecule that specifically binds the tetraplex assembly. Over the years, a small number of antibodies and single-chain variable fragments (scFv) have been used and validated to detect G-quadruplexes directly in cells. The latter correspond to structures where the heavy-chain and light-chain variable portions of a traditional antibody, which, when combined, form the antigen recognition surface, have been synthesized into a single polypeptide chain. Among these, the probes Sty3 and Sty49 were originally selected by ribosome display from the Human Combinatorial Antibody Library for high affinity binding of the ciliate *Stylonychia lemnae* telomeric repeat d(G_4_(T_4_G_4_)_4_) and provided the first evidence of G-quadruplexes in isolated nuclei [126]. Sty3 was found to specifically recognize parallel G-quadruplex structures at picomolar affinity [126], while Sty49 showed comparable nanomolar affinity for both parallel and antiparallel conformations [126]. Probe binding was validated in vitro by in situ immunofluorescence on isolated *S. lemnae* macronuclei, where Sty3 produced no signal, while Sty49 binding was shown to be specific for folded G-quadruplexes [126,127]. Importantly, Sty49 immunostaining provided the first direct evidence that G-quadruplexes are absent from replicating telomeres, which was expected due to the tetraplex assembly potentially inducing polymerase arrest during replication [126]. It is important to note that these studies were performed on isolated macronuclei from hypotrichous ciliates, where telomeric DNA concentration is far higher compared to other species [128,129]. Two other single-chain variable fragments were validated by immunofluorescence experiments on human cells, proving to be appropriate solutions to explore G-quadruplex biology in metazoans. BG4 was isolated by phage display and confirmed to bind various G-quadruplex conformations in vitro [130]. BG4 staining in multiple human cancer and non-cancer cell lines shows mostly nuclear foci, with fluorescent signals increasing upon G-quadruplex stabilization induced by PDS treatment [130] and being abrogated in DNase I-treated or G4-folded oligos-transfected cells [130]. Remarkably, it was revealed that BG4 foci mostly do not colocalize with telomeres and that their number peaks in S phase, in accordance with the hypothesis that ssDNA exposure during replication favors G-quadruplex formation [130]. In addition, BG4 immunostaining was proven to detect cytoplasmic RNA G-quadruplexes [131]. Although the original study showed that BG4 could bind parallel, antiparallel, and mixed propeller G-quadruplexes in vitro [130], an independent group demonstrated that the single-chain antibody preferentially binds parallel conformations by EMSA experiments [132]. A less commonly employed ScFv designed to bind to G-quadruplexes is D1. Like BG4, D1 was identified by phage display and it was specifically chosen for its marked binding preference to parallel G-quadruplex structures in ELISA assays, as well as for its clear BG4-like foci signals in immunofluorescence [133]. Co-staining with the telomere binding protein TRF2 in human SiHa cells showed that telomeric repeats were mostly folded in parallel G-quadruplex assemblies in vivo [133]. In addition to BG4 and D1, the traditional monoclonal antibody 1H6 was generated using vertebrate and ciliate telomeric G-quadruplexes as immunogens [134]. 1H6 was shown to recognize most G-quadruplex conformations in vitro and to specifically bind folded tetraplexes in immunofluorescence and immunohistochemistry experiments, with an increase in fluorescent signals upon treatment with the G4 stabilizer TMPyP4 or knockout of G-quadruplex helicase FANCJ [134]. Despite the possibility of performing isotype controls in experiments, 1H6 was later found to exhibit cross-reactivity with adjacent thymines in both folded G-quadruplexes and denatured DNA while not recognizing G-quadruplex structures bearing less than three adjacent thymines [135], thus complicating the interpretation of 1H6 experiments and casting significant doubt on the results obtained with this tool [136]. Overall, owing to 1H6 cross-reactivity and D1 specificity for parallel G-quadruplexes, BG4 has been the most frequently used G-quadruplex probe used to assay multiple tetraplex conformations at the same time. This does not exclude the possibility that BG4 may exhibit a bias towards some G-quadruplex folds over others; thus, particular care must be taken in not generalizing results. Despite this, BG4 has been used to adapt several other protocols for G-quadruplex studies in cells. Folded G-quadruplexes in cells were previously assayed indirectly by ChIP-seq of proteins known to recognize such structures [114,137,138,139]. Direct evidence of genomic loci bearing folded tetraplexes in native chromatin was obtained by BG4 ChIP-seq, which showed that only a small fraction of the quadruplexes observed in vitro are present in vivo in human cell lines and that the pattern of folded G-quadruplexes is cell-line- and cell-state-specific [85,140,141]. The original protocol was based on BG4 incubation after chromatin extraction [140] but it was subsequently combined with the CUT&TAG technology to allow in situ BG4 binding and to increase the signal-to-noise ratio [142,143]. Additionally, while both assay methods folded G-quadruplexes across a population of cells, BG4 CUT&TAG was adapted to capture genomic location of guanine tetraplex at the single cell level [144], thus allowing unprecedented insight into G-quadruplex heterogeneity in a cell population. The advantage of the adaptability and good performance of BG4 in these assays is counterbalanced by the fact that only this specific single-chain variable fragment has been widely used to study G-quadruplex prevalence in chromatin. In addition to concerns over BG4 preferential binding to certain G-quadruplex conformations over others, it is important to note that BG4 specificity is yet to be thoroughly tested. Usually, the specificity of an antibody for a ChIP experiment is evaluated in cells depleted of the protein of interest by knockout or knockdown approaches [145], which cannot really be realized with DNA secondary structures such as G4s [1]. Moreover, G-quadruplexes recognized by BG4 in crosslinked chromatin might not correspond to those actually found in live cells, skewing BG4 ChIP results even further from in vivo condition and favoring CUT&TAG-based approaches. To bypass this problem, expression of the G-quadruplex specific probe directly in cells before formaldehyde treatment should be performed. Unfortunately, this could be difficult with BG4, given that the reducing conditions in cytosol and the nucleus prevent the formation of key disulfide bonds necessary for proper scFv functionality [146,147,148]. On the other hand, the D1 probe was successfully expressed by transfected SiHa cells to perform ChIP-seq [133]. While D1 principally binds parallel G-quadruplexes, a recent study reported the production of a G-quadruplex-specific nanobody that was expressed in live cells to assay several G-quadruplex conformers in native chromatin by CUT&TAG [149]. Even though they are still far from perfect, the use of these probes, together with the more established BG4, should yield a less biased picture of G-quadruplex prevalence in cellulo. 

### 3.4. Small-Molecule G4 Stabilizers and Destabilizers

Another fundamental tool to study G-quadruplexes in their native environment is based on small-molecule ligands. Historically, the search for compounds capable of binding to guanine tetraplexes had been linked to their possible use in anticancer therapy, given that telomerase activity was found to be inhibited by G-quadruplexes [13]. Indeed, the first G4 ligand to be reported in the literature, 2,6-diamidoanthraquinone, was shown to inhibit telomerase through stabilization of G-quadruplexes [150]. Since that discovery, the number of compounds either specifically designed to bind G-quadruplexes or repurposed from other fields has grown substantially (for some noteworthy examples, see Figure 3). Currently, about 3700 molecules are listed in the G4LDB 2.2 database as G-quadruplex binders [151,152], forming a structurally heterogeneous group of compounds that mostly stabilize these tetraplex structures. According to a recent review [153], organic G4 ligands exhibit multiple planar aromatic rings arranged into three main types of architectures: (a) fused aromatic polycyclic systems, (b) macrocycles, and (c) non-fused aromatic systems, mostly populated by modular G-quadruplex ligands. The rationale behind most ligand designs is the generation of a large planar surface capable of establishing π-π stacking with the external G-tetrads of the target (as exemplified by the NMR-derived structure of the drug RHPS4 in complex with a parallel G-quadruplex [154]), together with cationic groups that form hydrogen bonds directly with the grooves and loops of the tetraplex assembly [155]. Different rules have been established for metallo-organic complexes, a rapidly expanding group of G-quadruplex-binding compounds reviewed in ref. [156]. Early chemistry studies had the explicit objective of developing G4 ligands that could function as antitumor drugs rather than simply studying these structures. As an example, in the 2000s, the trisubstituted acridine derivative BRACO-19 was shown to selectively recognize G-quadruplexes and inhibit telomerase in in vitro assays, achieving limited cytotoxicity in cancer cell lines, partial tumor xenograft regression in nude mice, and inhibition of the HIV-1 reverse transcription process [157,158,159,160]. Similarly, the cationic porphyrin TMPyP4 exhibited low cytotoxicity while inhibiting telomerase in several human cancer cell lines [161]. TMPyP4-mediated stabilization of G-quadruplexes in vitro occurred at slightly different affinities depending on tetraplex topology [162], although its selectivity towards quadruplex DNA was challenged in some studies [163,164]. Nevertheless, cell lines treated with the drug showed transcriptional downregulation of *c-myc* and other downstream genes in a manner dependent on G-quadruplex stabilization [165,166]. Another family of G-quadruplex-binding compounds is that of bisquinolinum derivatives, with its widely used member Phen-DC3. Its selective binding to G-quadruplexes in vitro was apparently similar to that of previously mentioned compounds and no changes in affinity to different G-quadruplex conformations were reported [124,167,168]. Interestingly, Phen-DC3 treatment of HeLa cells resulted in transcriptional changes in several genes bearing at least one sequence predicted to fold into a G-quadruplex [169], suggesting that the ligand could stabilize these secondary structures in vivo. An additional example of G4 ligand widely used in the literature is pyridostatin (PDS). After confirming its selectivity for human telomeric G-quadruplexes in vitro, PDS was shown to cause dissociation of POT1, a component of the human shelterin complex, from duplex telomeric DNA [170]. Moreover, the compound was demonstrated to induce DNA damage in cells at genomic sites enriched for putative G-quadruplex-forming sequences, to reduce the expression of genes close to DNA damage sites, and to inhibit the growth of several cancer cell lines [171,172]. 

A key advantage of small-molecule G-quadruplex ligands is the possibility of using them on live cells to explore the effects of tetraplex stabilization or to just exploit their ability to recognize folded G4s in native chromatin. Several compounds have been modified or purposefully designed to function as fluorescent probes in cells, establishing a new way to image G-quadruplexes without fixing cells and reducing the likelihood of preventing tetraplex-interacting proteins from binding the structure. For example, DAOTA-M2 has been recently described as a fluorophore capable of binding G-quadruplex in live and fixed cells [175]. G-quadruplex dynamics with this probe was assessed by Fluorescence Lifetime Imaging Microscopy (FLIM) with limited cytotoxic effects [175]. Similarly, the pyridostatin derivative SiR-PyPDS has been synthesized by tethering the fluorophore silicon-rhodamine (SiR) to the PyPDS scaffold [174]. This probe showed specific binding to several G-quadruplex oligonucleotides without significantly inducing G-quadruplex folding [174], a condition that would have altered tetraplexes in cells and complicate interpretation of results. Low nanomolar concentrations of SiR-PyPDS were used to perform single-molecule imaging in live cells, allowing scientists to study folding and unfolding events of single G-quadruplexes in situ [174]. Alternatively, click chemistry on G4 ligands already inside cells can be used to generate a fluorescent probe. A derivative of Phen-DC3 was labeled with the fluorophore Cy5 in fixed cells to show drug accumulation in the nucleus [176], while a similar feat was accomplished by linking a pyridostatin scaffold to an Alexafluor azide moiety [171]. A similar strategy was exploited to capture G-quadruplex-interacting proteins in cellulo. Two studies reported a screening approach where a small tetraplex ligand is photo-crosslinked to nearby proteins which are later identified by mass spectrometry analysis [177,178]. These techniques have proven successful in identifying new candidate proteins that modulate G-quadruplexes in native chromatin, expanding the current view of how cells regulate these structures [177,178]. Moreover, the same approach can be used without much modification by changing the G-quadruplex ligand, thus allowing researchers to reduce bias and to target specific nucleic acid structures or a subset of their conformations. Indeed, thanks to the availability of numerous G-quadruplex structures showing different conformations and topologies, small molecules can be designed and optimized for selective recognition of one or few of the known structures, enabling modulation and investigation of only tetraplexes with a specific conformation to provide insight into the biological relevance of G-quadruplex polymorphism. This strategy was successfully applied for various compounds, although target binding for most of them was evaluated only in vitro [54,55,179,180,181]. A successful example is IZCZ-3, a carbazole/imidazole derivative that specifically binds a parallel *c-myc* G-quadruplex over other tetraplex conformations [182]. The compound was shown to downregulate *c-myc* transcription through stabilization of its G-quadruplex in cells and to inhibit cancer cell growth by inducing cell cycle arrest [182]. On the other hand, optimization of the pyridostatin scaffold led to the synthesis of carboxyPDS, which was shown to selectively bind RNA G-quadruplexes over DNA tetraplexes in vitro and in cellulo [131,183], allowing precise investigation of the biological role of RNA tetraplexes. Overall, chemical biology methods to study G-quadruplexes have flourished in the past two decades thanks to an increase in the available G-quadruplex structures. Time and time again, small-molecule ligands have proven fundamental in investigating G-quadruplex prevalence and dynamics in live cells, which would have been impossible to do with antibody-based methods. Additionally, the modularity of certain scaffolds, the possibility of performing click chemistry in situ, and the rational design and optimization of these probes allowed the synthesis of thousands of molecules, some specifically targeting certain G-quadruplex conformations over others. This vast collection of compounds was shown to stabilize guanine tetraplexes in vitro and in cells, providing new insight into the effects of altered G-quadruplex dynamics. Considerable effort has gone into the analysis and optimization of G-quadruplex stabilizing probes, thus leaving the interesting avenue of G4-destabilizing compounds mostly unexplored. To date, only a few drugs have been shown to alter and unfold G-quadruplexes; however, the conclusions drawn from these experiments are still debated (reviewed in ref. [184,185]). Interestingly, one such compound was shown to upregulate *c-kit* transcription [186], which was previously shown to be reduced upon G-quadruplex stabilization [187]. In light of the possibility of investigating the effects of reduced G-quadruplex levels, more effort should be devoted to the discovery and synthesis of these compounds. 

## 4. Prevalence of Guanine Tetraplexes in Genomes

After having discovered G-quadruplexes and probed their structure in numerous studies, the next logical step consisted of understanding where they are found in genomes and if their position has with a possible function. As explored previously, the most straightforward method used to identify G-quadruplexes in genomes is based on bioinformatic prediction. By virtue of knowing the structural requirements of G-quadruplex formation, it is possible to identify which sequences have the propensity to fold into G-quadruplexes and identify them in a genome of interest. Even the most conservative algorithms for putative G-quadruplex-forming sequence (PQS) yielded more than 375,000 hits in the human genome [110,111], with more recent strategies taking into account non-canonical G-quadruplex structures and machine learning to improve sensitivity [112], thus increasing the number of genomic sites that could potentially fold into canonical and non-canonical G-quadruplexes. Furthermore, the combination of bioinformatic prediction and high-throughput polymerase stop assays coupled to next generation sequencing [121,123] greatly expanded the view of G-quadruplex-forming sites in the human genome, revealing that transcription start sites, 5′UTR, and splicing regions have the highest enrichment of these structures, while comparatively few of these structures are found in coding sequences [121,188,189]. The rG4-seq dataset on the human transcriptome revealed that G-quadruplexes in mRNA have the highest density in 5′ and 3′ UTRs compared to CDS (in accordance with G4-seq data) [123].

Importantly, the same strategy was applied to various genomes across diverse species to provide insight into taxa-specific characteristics and the evolution of G4 motifs. Putative G-quadruplexes were found to be enriched in gene regulatory regions of several prokaryotic genomes, indicating a role in transcriptional regulation of specific gene classes [190,191,192,193,194,195,196,197]. For example, this link was confirmed for the specific case of radioresistance genes in *Deinococcus radiodurans*, whereby upon treatment with the small-molecule G4 ligand N-methyl mesoporphyrin (NMM), a drop in target gene transcription and in resistance to radiation was observed [198]. Bioinformatic analysis of human viral genomes revealed that putative G-quadruplex sequences were found in most virus categories, particularly in single-stranded genome viruses, with the dsDNA *Herpesviridae* family being a notable exception [199]. An example of the functional relevance of G-quadruplexes in viral genomes was provided in a reporter assay of PQS-containing human cytomegalovirus (HCMV) regulatory sequences, where chemically induced G-quadruplex stabilization resulted in transcriptional suppression of some of the tested viral promoters [200], suggesting that these secondary structures can be used as targets for specific antiviral drugs (reviewed in ref. [201,202]). Interestingly, a fraction of the predicted G4-forming sequences from the unicellular eukaryote *Saccharomyces cerevisiae* was found to be conserved in other fungal species [203], with these sequences being enriched in promoters, telomeres, rDNA, mtDNA, and sites of frequent double-strand breaks [203,204]. In contrast, among several plant species, the prevalence of potential G-quadruplex-forming motifs shows great variability, with the highest density observed in monocots and lycophytes compared to dicots and bryophytes (although this could be due to the difference in GC content between plant genomes) [205]. Moreover, G4 motifs with stretches of two guanines are strikingly more abundant than the standard G_3_ sequences, and putative G-quadruplex location dramatically favors intergenic regions over coding regions [205,206]. Despite this, about a fifth of annotated *Arabidopsis* genes harbor at least a putative G-quadruplex in their promoter, as well as 68% of gene models (i.e., all possible ORFs, including splice variants) [206], while in *Zea mays*, enrichment of G4 motifs was found to be particularly high in the proximity of the transcription start site [207]. Thus, the possibility of G-quadruplexes regulating gene expression at the transcriptional or post-transcriptional levels in plants cannot be excluded. Similarly, several metazoan species reportedly show a significant proportion of annotated genes with putative G-quadruplex-forming motifs found in the 2 kb upstream region [208], although the gene fraction in representative genomes from warm-blooded animals was much higher than that from cold-blooded animals [209]. A more dramatic difference between these animal groups was also reported when considering the frequency of predicted G-quadruplexes in a 1000 bp window around the transcription start site [209]. These findings were later confirmed and expanded upon using an optimized G4-seq protocol, which showed that sequences that can fold into G-quadruplexes in near physiological conditions are more common in promoters and 5′UTRs in human, mice, and *Trypanosoma* genomes, while these observed G-quadruplexes are not significantly enriched in *C. elegans*, *D. rerio*, or *D. melanogaster* [122]. Overall, pattern-matching algorithms for putative G-quadruplex searches revealed that these structures are widespread across evolutionarily distant genomes, in all domains of life and viral species alike. These studies show that the overall G-quadruplex density between genic and intergenic regions changes significantly across taxa [122], as does the G4 motif, particularly in loop length and variability of the core sequence [122,205,208]. Despite this, multiple reports underline that putative G-quadruplexes, as well as assemblies validated by G4-seq, are present at a significant level in gene regulatory regions, although the extent of this enrichment is extremely variable across taxa [122,190,199,204,205,208,209]. Importantly, one study [208] found that the location of putative G4 sequences shifted from evenly distributed to clustered at different chromosomal loci from basal eukaryotic species to higher organisms such as humans and *Gallus gallus*. Moreover, G4 motifs within the 2 kb regulatory region upstream of genes tended to accumulate across evolution in loci coding for transcription factors. This suggests that, during the evolution of eukaryotic species, G-quadruplexes gradually acquired a relevant function in gene expression regulation, particularly at the transcriptional level, and, therefore, G-quadruplex-forming sequences were slowly concentrated at strategic regulatory regions close to genes. The fact that genes coding for transcription factors show an abundance of putative G-quadruplex sequences [208] would then explain why G-quadruplexes have been linked to the wide variety of biological functions [208], which will be discussed in a later section. In the literature, various cases of specific G4 loci that are phylogenetically conserved have been reported. To report some examples, a validated G4 locus in the *RPB1* gene coding for the large subunit of RNA polymerase II has been found to be conserved in the Archaeplastida plant supergroup [210]. Additionally, a G-quadruplex-forming sequence was found in the 5′UTR of the *NRAS* proto-oncogene’s mRNA in at least six different mammalian species, where it may have a role in translational regulation [211]. To sum up, G-quadruplexes are a feature common to all genomes analyzed so far, irrespective of species complexity. Moreover, these structures do not appear to have been selected against during evolution, but rather were progressively co-opted, likely to serve as a new layer of gene expression regulation. 

## 5. Recognition and Modulation of G-Quadruplexes in Cells

As mentioned previously, only a small fraction of all sequences capable of folding into G-quadruplexes actually forms tetraplexes in vivo [85]. Indeed, the cell type and differentiation state are correlated to a different G4 repertoire [140,141]. In addition, since G-quadruplexes are not under negative evolutionary pressure to be eliminated, they have likely been domesticated to perform some kind of biological function [208]. This implies that the cell must be able to recognize these structures and induce their folding or unfolding when and where needed. G-quadruplex polymorphism may thus be biologically relevant, meaning that cells would be capable of discriminating between different tetraplex topologies and conformations. It is therefore not surprising that a large number of proteins have been discovered that are capable of G-quadruplex binding.

### 5.1. Approaches to Identify G-Quadruplex-Binding Proteins

Among the several strategies that have been used for the identification of tetraplex-binding proteins, the most common relies on affinity pull-down of cell extracts using DNA or RNA G-quadruplex oligonucleotides [212,213,214]. This approach heavily depends on the type of bait being used and cannot capture protein-binding events that are modulated by flanking sequences or the chromatin context. These issues have been partially addressed with the development of co-binding mediated protein profiling assays, where proteins interacting with G-quadruplexes in native chromatin are identified through photo-crosslinking to a G4 specific probe [177,178]. On the other hand, an alternative indirect method consists of identifying proteins whose binding sites, assessed by ChIP, contain a putative G-quadruplex forming sequence [137,215]. However, additional assays are required to validate the actual binding to folded G-quadruplexes rather than the corresponding dsDNA sequence. A third type of screening is based on bioinformatic analysis of protein sequences from a wide collection of previously identified tetraplex interacting proteins to reveal common motifs. A recent study highlighted the enrichment of RGG motifs in G4-binding proteins [216], which was shown to mediate interaction with the nucleic acid structure through hydrogen bonding and π-stacking provided by key arginine and phenylalanine residues [217]. The authors propose that putative G-quadruplex-binding proteins could be more rapidly identified through the presence of the RGG domain in their sequence. Although compelling, this approach heavily relies on a specific protein domain to predict G-quadruplex binding of uncharacterized proteins. Considering the current state of G-quadruplex–protein interaction studies, it is safe to assume that new domains and motifs will be identified as mediators of tetraplex binding in the future, leading to less biased predictions. Nevertheless, candidate protein–G4 binding must be confirmed in in vitro assays, while the biological relevance of the interaction should be explored via protein depletion or overexpression in cell lines or other biological systems. 

### 5.2. Established G-Quadruplex-Binding Proteins

As of 26 February 2024, the G4IPD database for G-quadruplex interacting proteins [218] (available at https://iiti.ac.in/people/~amitk/bsbe/ipdb/index.php, accessed on 26 February 2024) lists about 60 and 40 entries that recognize DNA and RNA G-quadruplexes, respectively. This collection of factors was demonstrated to either stabilize or destabilize the tetraplex structure (reviewed in ref. [219] and visually grouped in Figure 4). The latter is mainly performed by G-quadruplex-unwinding helicases (reviewed in ref. [220]). Among these, the mammalian RecQ helicases WRN and BLM have been studied for more than two decades. Both proteins were shown to unwind both dsDNA substrates, as well as G-quadruplex forming oligos in vitro in 3′ to 5′ directionality and in the presence of ATP, provided they have a centrally located ssDNA region and a short 3′ ssDNA overhang [221,222,223,224]. In a single molecule FRET assay, the mechanism of BLM binding to the substrate was found to involve both the RecQ C-terminal (RQC) and Helicase-RNase D C-terminal (HRDC) domains, with the latter recognizing the ssDNA region and aiding RQC binding and unfolding of the G-quadruplex in an ATP-independent manner [224]. In another study using smFRET, it was demonstrated that BLM could recognize and unwind substrates regardless of tetraplex conformation, while WRN was shown to selectively destabilize only telomeric G-quadruplex substrates [53]. Helicases are not only involved in modulation of genomic G-quadruplexes, but also process RNA tetraplexes (reviewed in ref. [225]), as in the case of DEAH box protein 36 (DHX36). Also known as RNA helicase associated with AU-rich element (RHAU) and G4 Resolvase 1 (G4R1), the helicase unwinds both RNA and DNA G4s in vitro [226]. It was found to selectively recognize parallel tetraplex structures through its short N-terminal domain interacting with the exposed hydrophobic surface of a terminal G-tetrad and electrostatic interactions with backbone phosphate groups [227,228]. Mechanistic analysis of DHX36 activity revealed that it requires loading on a 3′ ssDNA overhang and then progressively unwinds substrate G4s in a 3′ to 5′ directionality through ATP hydrolysis, similar to other DEAH-box helicases [229]. Overall, BLM, WRN, DHX36, and all the other G-quadruplex helicases have been found to destabilize guanine tetraplexes in vitro and in vivo, where the unwinding activity has been linked to DNA replication and gene expression regulation. Indeed, it has long been hypothesized that G-quadruplex formation in the template strand would induce DNA or RNA polymerase stalling, leading to downregulated gene expression [230,231,232] and genome instability [233,234,235]. These consequences will be discussed extensively in following sections.

Besides helicases, other proteins participate in G-quadruplex binding and regulation. A curious example is nucleolin, a multifunctional phosphoprotein commonly found in nucleoli that was shown to bind and stabilize G-quadruplexes in the *c-myc* upstream regulatory element NHEIII_1_ [236]. Interestingly, nucleolin binding in vitro was shown to be dependent on the presence of at least one long loop (more than 3 nt) in the structure, independently of the underlying sequence or tetraplex conformation [52]. This suggests that the chaperone activity of nucleolin towards G-quadruplex-forming sequences is targeted mainly or exclusively at those loci where these long loops are formed [236]. Moreover, nucleolin-mediated stabilization of G-quadruplexes was reported to stimulate or inhibit transcription of *c-myc* in vitro [236,237], and VEGF in vitro and in vivo [238], two genes containing guanine tetraplexes in their promoters. Nucleolin is not alone in stabilizing G-quadruplexes; LARK and apurinic/apyrimidinic endonuclease 1 (APE1) were found to participate in G-quadruplex folding and stability, a function that was closely tied to transcriptional regulation [239,240]. Besides transcriptional modulation, G-quadruplex stabilization is involved in telomere DNA maintenance. 

Not all G-quadruplex-binding proteins mainly function to regulate the stability of these secondary structures. An increasing proportion of these are recruited at G-quadruplex-forming sites through direct binding to the tetraplex to perform their molecular functions on other proteins or on nearby nucleic acids. For example, the telomeric repeat binding protein TRF2 was shown to bind simultaneously telomeric dsDNA and tetraplex telomeric RNA TERRA in vitro, opening the possibility of TRF2 functioning in vivo as glue between the two and mediating TERRA localization in telomeric DNA [241]. Moreover, in vitro binding assays showed that three human DNA methyltransferases (DNMT1, DNMT3A, and DNMT3B) could selectively bind to G-quadruplexes [242], while DNMT1 occupancy at chromatin sites in K562 cells was revealed to be enriched at hypomethylated G-quadruplex forming sites, as evaluated by BG4 ChIP-seq [243]. Based on these results, the authors proposed that folded G-quadruplexes in chromosomal DNA sequester DNMT1, inhibiting its activity and protecting nearby sequences from methylation [243]. While this inhibitory mechanism is yet to be demonstrated in vivo, this compelling evidence links guanine tetraplexes to epigenetic regulation through protein recruitment at folded G-quadruplexes in the genome, a topic that will be discussed further in a later paragraph.

To summarize, a wide variety of proteins has been shown to physically interact with G-quadruplexes, both in vitro and in vivo. A large fraction of G-quadruplex-interacting proteins identified so far regulate the stability of the structure by actively unwinding it or promoting its folding, showing that the cell is capable of regulating G-quadruplex dynamics according to its needs. Interestingly, recent studies have revealed the existence of proteins that function as G-quadruplex readers, in that they bind selectively to these folded structures to concentrate their activity at specific genomic regions. The cell’s ability to modulate and recognize its guanine tetraplexes in a conformation- and topology-specific manner confirms that these are bona fide structures in nucleic acids. It is important to note that a deeper understanding of G-quadruplex biology relies on the identification and characterization of new G-quadruplex-interacting proteins. Several proteome screening methods have been developed to expand the list of these proteins, each with its own advantages and drawbacks. As new G4-interacting proteins are identified and studied, our understanding of G-quadruplex biology will inevitably change. As of now, hypotheses and models on the functions of tetraplex assemblies inevitably paint an incomplete picture, which is, however, constantly and rapidly updated as more research is performed in this area. 

## 6. Physiological and Pathological Roles of G-Quadruplexes

### 6.1. G-Quadruplexes in Transcriptional, Post-Transcriptional, and Epigenetic Regulation

Having established that G-quadruplexes are biologically relevant, the main functions they are involved in will be presented and are recapitulated in Figure 5. One of the most extensively studied areas in this context is transcription. As early as the start of the millennium, G-quadruplex involvement in transcriptional regulation of oncogenes was identified [166]. Additionally, as presented previously, the first computational predictions of guanine tetraplexes from newly sequenced genomes revealed that these putative G4-forming sequences were abundant in gene promoters [110,111]. This feature was confirmed in BG4 ChIP-seq experiments [85,141,244], further strengthening the link between guanine tetraplexes and transcription. The original molecular model considered folded G-quadruplexes simply as obstacles to RNA polymerases, thus causing a reduction in transcript levels. This idea was built on data from in vitro transcription assays showing RNA polymerase arrest when elongating G-rich or G-quadruplex-forming templates [245,246]. Although this model was reinforced by the demonstration that genes bearing G-quadruplexes near their transcription start sites were downregulated upon G4 stabilization by small-molecule ligands in cells [165,166,187,247], this could be the result of indirect consequences of global guanine tetraplex stabilization [248,249]. Over the years, the role of G-quadruplexes in transcription has been expanded greatly, painting a more nuanced picture of where G-quadruplexes exert transcriptionally stimulating or silencing effects in a context-dependent manner. A key finding was the characterization of transcription factors that bind folded G-quadruplexes and thus could be recruited at certain genomic sites [215,250,251,252]. Moreover, folded G4s as detected by BG4 ChIP-seq in HaCaT cells were almost exclusively found in nucleosome-depleted regions of genomic DNA [85], a chromatin feature that is common at eukaryotic transcription start sites and that more broadly modulates DNA accessibility to transcription factors (reviewed in ref. [253]). Although a clear causal link has yet to be established, it is possible that G-quadruplex formation at selected genomic sites prevents nucleosome assembly, hinting at another mechanism of gene expression regulation by these secondary structures [254]. Overall, while G-quadruplex involvement in all stages of transcription has been extensively reported and several potentially compatible models of the underlying mechanisms have been proposed, solid evidence of direct modulation of chromatin states by guanine tetraplexes is lacking. Nevertheless, exploration of the transcriptional role of these structures is currently underway, especially considering the clinical importance of this mechanism.

The role of G-quadruplexes in gene expression regulation is not limited to transcription, as evidence of their involvement in RNA biology and translation has been reported [2,255]. Although mainly studied in the context of mRNA, G-quadruplexes have also been detected in diverse classes of non-coding RNAs, such as lncRNAs [256], pri-miRNAs [257], and piRNAs [258]. Guanine tetraplex assemblies have been reported to play an important role in pre-mRNA maturation, particularly in the context of alternative splicing, as shown for FM1, p53, and hTERT primary transcripts [259,260,261,262]. It has been demonstrated that G-quadruplexes recruit splicing factors, such as hnRNPH [263] and hnRNPF [264], and that global destabilization of RNA guanine tetraplexes results in widespread splicing deregulation [265]. Besides mRNA maturation, its positioning within the cell is also regulated by G4s, for example, functioning as a signal for neurite targeting of selected transcripts [266]. Another intriguing example is their involvement in phase separation, as structured G-quadruplexes in C9ORF72 mRNA were found to induce stress granule assembly [267]. Similarly, a recent study by Wang and Xu demonstrated that short RNA molecules bearing CGG or telomeric repeats aggregate into solid state foci, both in vitro and in vivo, through the formation of intermolecular higher-order G-quadruplex structures [268]. Furthermore, RNA G-quadruplexes have been shown to have a direct effect on translation itself, for example, by preventing recruitment of the 43S ribosome and reducing translation efficiency unless unwound by the translation initiation factor eIF4A [269]. Another mechanism that has been reported in the literature is translational repression of mRNAs harboring rG4s that induce ribosome positioning at uORFs [270]. In one of the first instances where this phenomenon was observed, the RNA G-quadruplex in the 5′UTR of *NRAS* was shown to inhibit translation of a reporter gene in vitro [211]. To generalize, tetraplex assemblies in 5′UTRs of mature transcripts have been shown to modulate translation rates, albeit in a context-dependent manner rather than based solely on sequence or 3D structure [271]. 

Another layer to gene expression regulation where G-quadruplexes have been implicated is in chromatin organization and epigenetics (reviewed in ref. [6]). As discussed previously, G4s are enriched in nucleosome-depleted regions of transcriptionally active genes, suggesting a role in nucleosome positioning. This concept has been explored in the context of epigenetic heritability during DNA replication, where chemical G-quadruplex stabilization was tied to loss of H3K4me3 and subsequently addition of methylated cytosines. This epigenetic reprogramming was maintained throughout cell generations and was shown to directly repress transcription of nearby genes [272]. Non-replicative histone deposition is also influenced by G-quadruplexes through recruitment of histone chaperones, as has been demonstrated for ATRX-mediated deposition of H3.3 at telomeres [114,273] and the removal of the H2A-H2B dimer performed by G-quadruplex-binding nucleolin [274]. Moreover, histone post-translational modifications can be modulated in a G-quadruplex-dependent manner through the recruitment of histone modifying enzymes at sites of folded G-quadruplexes. Besides the cases of DNMT1, 3A, and 3B [242,243], which were explored in a previous section, this has additionally been shown by G4-mediated recruitment of the LSD1–CoREST complex at the hTERT gene promoter, where it catalyzes the demethylation of H3K4/9 [275]. On the whole, multiple studies have directly or indirectly linked folded G-quadruplexes to chromatin remodeling and epigenetic factor recruitment to affect local chromatin structure. In recent years, it has been suggested that the concept of guanine tetraplexes as protein recruitment hubs could also be expanded to include regulation of long-distance chromatin contacts, particularly pertaining to distal promoter–enhancer interactions [249]. The validity of this hypothesis was strengthened when the combination of BG4 ChIP-seq and chromatin conformation capture data revealed an enrichment of folded G-quadruplexes at topologically associated domain (TAD) boundaries [276], which correspond to regions of gDNA that are distant in linear sequence but are found to be closely associated in the three-dimensional organization of chromatin thanks to the DNA looping factors cohesin and CTCF [277,278]. Furthermore, binding sites of several architectural proteins were enriched in G-quadruplex-containing TAD boundaries, pointing to G-quadruplex-dependent loop formation via their recruitment [276]. These correlations were confirmed by Li and colleagues in a 2021 paper, where the transcription factor and DNA looping protein YY1 was shown to bind G-quadruplexes in cells and dimerize to generate long-distance chromatin contacts partly due to guanine tetraplexes to then mediate gene expression regulation [279]. 

Overall, G-quadruplexes have been implicated in both transcriptional and post-transcriptional regulation by a variety of proposed or demonstrated mechanisms, making the original claim that these structures work simply as physical blocks to RNA polymerase obsolete. While more data on the native context of these processes are needed to faithfully explore G4 involvement in gene expression regulation, it can be nonetheless stated that a critical though not exclusive component of their role relies on recruiting proteins (transcription factors, splicing factors, translation factors, etc.) in selected genomic or RNA regions. This highlights further that understanding G-quadruplex biology at a deep level necessarily requires the identification and characterization of G-quadruplex-binding proteins.

### 6.2. Impact of G-Quadruplexes on DNA Replication

As for other non-canonical secondary structures, the presence and dynamics of G4s in nucleic acids have a significant impact in the context of DNA integrity and replication (reviewed in ref. [280]). As mentioned earlier, dsDNA unwinding during transcription or replication exposes single-strand DNA, favoring G-quadruplex folding, which creates obstacles to DNA polymerase progression, as shown in in vitro polymerase stop assays [281]. The concept of guanine tetraplexes causing DNA polymerase stalling in vivo has been strongly hinted at in numerous studies, with the underlying molecular mechanisms and the strategies adopted by biological systems to prevent and resolve these blocks being hypothesized or experimentally proven. Evidence that folded guanine tetraplexes induce replication fork arrest in vivo has been described in various biological systems, ranging from *X. laevis* egg extracts [282] to *S. cerevisiae* yeast cells [139], a patient-derived cell line [234], and whole *C. elegans* animals [283]. The common observation of all these studies is that DNA replication is negatively affected by the presence of these secondary DNA structures. When cells are unable to unwind G-quadruplexes or resume replication fork progression, DNA deletions were observed, with sizes ranging from 60–300 nt (similar to Okazaki fragments) [283] to up to 80 kb [234]. In mechanistic explorations of this phenomenon, G-quadruplex-unwinding helicases are often implicated as critical factors in resolving the tetraplex and avoiding replication fork stalling [139,282]. Additionally, dedicated DNA polymerases are specifically recruited at the stalled fork to resume elongation downstream of the folded structure. This has been shown for REV1 [284] and PrimPol, the latter having G-quadruplex-binding ability in vitro and being able to reprime the template downstream of the guanine assembly [285]. However, when these factors are depleted or inactive, the G-quadruplex becomes an obstacle to replicative fork progression, leaving an exposed ssDNA stretch that eventually generates a double-strand break [286,287]. Similarly, chemical stabilization of G-quadruplexes using the ligand pyridostatin was correlated to induction of double-strand breaks as evaluated by neutral comet assay [171]. Moreover, it was reported that a single unresolved G4 remains stable throughout mitosis and is inherited by daughter cells, which, when undergoing S phase, use the same G4-containing strand as a template and thus are exposed to the same type of DNA lesion [286]. 

Having just explored the negative influence of folded G-quadruplexes on replication fork progression, it may be surprising to learn that tetraplex assemblies were shown to participate in replication origin firing in eukaryotes. Indeed, deep sequencing of short nascent strands (SNSs) in human cell lines revealed that active replication origins are enriched at transcription start sites of CpG-rich promoters. Importantly, most of the identified origins map to sites of putative G-quadruplex-forming sequences (PQS), with PQS density and origin efficiency showing a positive correlation [288]. Confirmation of a causal relationship was obtained by studying origin firing in a normally late-replicated locus where the PQS-containing chicken β^A^ origin was inserted. In this genetic background, the transplanted origin induced an enrichment in SNS signal while replication timing remained unaffected. Moreover, point mutations that destabilized the predicted G-quadruplex caused a drop in origin firing efficiency, directly tying folded tetraplexes to replicon activity. Importantly, G4 orientation in this system determined the direction of the replication initiation site [289], consistently with previous observations of SNS peak enrichment within 200 bp 3′ of a PQS [290]. Although these results established that G-quadruplexes regulate origin activity in metazoans, the molecular mechanism behind this phenomenon remains obscure. Currently, three main hypotheses have been proposed, using models of replication start sites derived from yeast studies as a foundation. Firstly, it has been observed that origins are enriched in nucleosome-depleted regions [291,292], a chromatin feature that has also been associated with G-quadruplexes [85,293] (although the causal relationship is yet to be defined). Another possibility lies in the fact that folded G-quadruplexes may function as recruitment hubs for factors involved in the initiation of replication. Indeed, the human Origin Recognition Complex (ORC) was found to specifically recognize tetraplex assemblies in RNA and ssDNA [294]. Furthermore, it is possible that G-quadruplex-unwinding helicases are recruited in folded tetraplexes near eukaryotic origins in order to remove the guanine assemblies and later unwind dsDNA to allow DNA replication to begin. This model would explain why folded G-quadruplexes determine the position of origin firing, since helicases have a defined unwinding directionality [280]. Overall, the idea that guanine tetraplexes play a role in replication origin activation and in determining the directionality of the initial dsDNA denaturation is intriguing. This phenomenon is in contrast with the simplistic notion that these secondary DNA structures are simply obstacles to DNA replication; instead, it highlights a more nuanced function in the wider context of genome maintenance. Further research is needed to elucidate how G-quadruplexes contribute to replication origin firing at the molecular level, particularly with regards to protein factors that are selectively recruited at those sequences.

### 6.3. G-Quadruplexes and the DNA Damage Response

Understanding how cells repair lesions caused by G-quadruplexes is important to comprehend and predict the consequences of impaired G4 removal. In general, the threat DNA lesions pose to genome integrity is avoided thanks to a complex system where the wide variety of repair pathways are specifically activated at damaged genomic sites through the DNA damage response (DDR) (reviewed in ref. [295,296,297,298]). In most models of DDR processes, the DNA lesion is represented as a signal that is detected by sensor proteins (upstream phosphorylation events), which then directly or indirectly transduce and amplify the signal to activate a wide variety of effectors to efficiently repair the damage or activate cell cycle arrest and apoptosis when that is not possible. The DDR signaling cascade is extremely complex in human cells; however, it is worth mentioning a few of its key players. Three phosphoinositide 3-kinase (PI3K)-related kinases (PIKKs) are principally involved in initiating the signaling cascade in eukaryotes: Ataxia-Telangiectasia Mutated protein (ATM), ATM- and Rad3-related protein (ATR), and DNA-dependent protein kinase (DNA-PK). Although acting in slightly different scenarios, all three apical kinases are specifically recruited at sites of DNA damage by the co-factors NBS1 (a MRN complex component), ATRIP, and Ku80, respectively. After autophosphorylation and other activating modifications, PIKKs trigger the signaling cascade by phosphorylating a wide range of targets, thus transmitting the signal to eventually activate terminal effectors. Among the earliest temporally, H2A.X phosphorylation at Ser139, principally mediated by ATM at DSB sites and referred to as γH2A.X, is a robust marker for DNA damage in cells [299,300]. H2A.X is a variant of the canonical core histone H2A that is central to most DDR pathways in that it sustains DDR signaling and spreads to nearby chromatin to organize an efficient response. In the literature, γH2A.X has often been described as a recruitment hub for several DNA damage-response proteins, as has been shown for BRCA1, P53BP1 and MDC1 [301,302,303]. This serves as both a mechanism to locally increase the concentration of repair factors at sites of DNA lesions and tether broken DNA ends together, as well as for generating a positive feedback loop to further enhance DDR signaling. Moreover, the response is articulated so as to activate cell cycle checkpoints and stop its progression. ATR and ATM phosphorylate the checkpoint kinases Chk1 and Chk2, respectively [304,305]. Both effector kinases target the protein phosphatase Cdc25A and induce its degradation, thus preventing removal of inhibitory phosphates from CDK2 [306,307]. Crucially, both ATM and ATR were shown to directly or indirectly stabilize the oncosuppressor p53 [308,309,310,311], which then upregulates genes involved in cell cycle checkpoint activation and apoptosis. Overall, through activation of apical kinases, DDR signaling triggers a multitude of parallel and partially redundant pathways to efficiently activate the appropriate DNA-repair pathway and coordinate a cell cycle arrest and apoptotic response to preserve genome integrity. The system is particularly robust since inactivation of a single pathway does not generally prevent the activation of many others with similar outcomes. However, an increased DNA lesion rate caused by the inability to remove obstacles to polymerases, such as stable G-quadruplexes, would increase the burden on this system, ultimately resulting in chromosome fragility.

In a 2014 paper, a dedicated DNA-repair pathway was implicated in the processing of damaged DNA resulting from failed G-quadruplex removal in *C. elegans*. According to the authors, the resulting double-strand break is repaired through single-nucleotide micro-homology and subsequent extension performed specifically by DNA polymerase θ, resulting in small deletions [287]. In addition, the presence of G4 structures can influence DNA-repair processes. For example, in the proximity of DSB breaks, these structures can inhibit ends processing, altering the balance between the Homologous Recombination (HR) and Non-Homologous End Joining (NHEJ) repair pathways [312,313]. Furthermore, G-quadruplexes promote genomic instability when combined with R-loops on the opposite DNA strand, forming a G-loop [75]. It has been shown that global G4 stabilization in a cancer cell line using two chemically distinct compounds leads to activation of the DNA damage response, as evaluated by H2A.X and ATM phosphorylation, as well as RAD51 and p53BP1 nuclear foci formation [314]. Interestingly, the same study reports a concomitant increase in R-loop levels 2 to 10 min after treatment with G4 stabilizers, showing that overexpression of RNase H1 in this system abrogates DNA damage induction as indicated by stable levels of γH2A.X [314]. Although the molecular mechanism behind G-loop-dependent DNA lesions has yet to be elucidated, the authors propose that the stabilizing effect G-quadruplexes have on R-loops could increase the risk of transcription–replication conflicts or of activating a recombination pathway that produces double-strand breaks [314,315]. Overall, guanine tetraplexes have been shown to block DNA (and RNA) polymerases in DNA replication and transcription, thus increasing the risk of damaging DNA [233]. Several factors have been discovered that prevent G4-mediated instability through tetraplex unwinding, repriming, or translesion synthesis. However, when one of these structures cannot be removed, DNA lesions are formed through a yet undefined molecular mechanism, therefore activating DNA damage-repair and recombination pathways. Research in this area is focused on precisely dissecting how G4-dependent DNA lesions are formed, as well as on analyzing the pathways that counteract genomic instability induced by guanine tetraplexes. Aberrant responses to tetraplex-induced DNA damage are particularly relevant in pathological conditions, a concept that will be explored later in this review. 

### 6.4. G-Quadruplexes in Telomeric Regions

It is noteworthy to consider that the earliest studies on G-quadruplexes were focused on telomeric sequences. Telomeric DNA is located at the ends of eukaryotic chromosomes and one of its defining features is the presence of short species-specific G-rich tandem repeats. Importantly, in order to prevent improper recombination events through chromosome termini, telomere ends are protected from DNA damage-repair proteins. In vertebrates, this is achieved through the formation of T-loop structures, consisting of the ssDNA 3′ overhang invading an upstream telomeric repeat dsDNA thanks to the shelterin protein complex [316]. Owing to the presence of repeats rich in guanines, several three-dimensional structures of telomeric G-quadruplexes have been reported in the literature. These studies originally used ciliate telomeric repeats, but have since been expanded to include human telomeric sequences [18,317,318]. Eventually, telomeric DNA G-quadruplexes were observed in cells thanks to BG4 and D1 immunostaining [130,133], prompting scientists to study the influence of these secondary structures on telomere biology. Indeed, it was found that a collection of telomeric proteins can recognize folded G-quadruplexes, namely, Rif1 [138] and the shelterin component Telomeric Repeat-binding Factor 2 (TRF2) [241]. When G-quadruplex impact on telomere end protection was explored, it was discovered that folded tetraplex assemblies provide weak telomere capping through recruitment of G-quadruplex-binding telomeric proteins [319]. In addition, DNA and RNA G4s were shown to participate in telomeric chromatin maintenance through recruitment of the human TLS/FUS protein, which then regulates local levels of H4K20me3 [320]. Besides their impact on telomeric chromatin structure, G-quadruplex levels modulate telomere lengthening processes. Part of the driving force behind research into G-quadruplex biology originated from early evidence that folded G-quadruplexes inhibit telomerase activity [13,150]. Importantly, the effect on telomerase is topology-specific; intramolecular antiparallel G-quadruplexes effectively block enzyme activity [13], while intermolecular parallel tetraplexes can be resolved and extended by telomerase [321]. Furthermore, as observed for non-telomeric DNA, unusually high levels of folded G-quadruplexes hamper DNA replication through telomeres. In fact, telomere shortening and fragility was observed in cells depleted from the G4 unwinding helicases BLM, WRN, and RTEL1 [322,323,324]. To further support the idea of G-quadruplexes as genome instability factors at telomeres, treatment with a collection of small-molecule G4 stabilizers results in similar phenotypes [325,326], which are further exacerbated when performed on cells with reduced G-quadruplex-unwinding capabilities [327,328,329]. Overall, folded G-quadruplexes have been shown to work as important structural elements in telomeres, possibly providing rudimentary telomere end capping and functioning as protein recruitment hubs to shape telomeric chromatin. Crucially, the three-dimensional structure of guanine tetraplexes determines their effect on telomerase, showing that structural polymorphism has biological significance. Furthermore, telomere replication and lengthening are negatively affected upon loss of regulation of G-quadruplex levels, inducing genome instability similarly to the roadblock model that was explored for DNA replication and transcription. 

### 6.5. G-Quadruplexes in Human Disease and Therapy

As just explored, G-quadruplex formation and dynamics have been connected to a number of cellular functions. It is therefore possible that abnormally high or low levels of guanine tetraplexes in a cell lead to deregulation of key cellular processes, a scenario that can be pathological if the effects exceed the tolerance of an organism. Indeed, a collection of disease-causing genetic mutations have been shown to increase the number and/or stability of G-quadruplexes at specific loci. A notable example is the GGGGCC tandem repeat expansion in the first intron of C9ORF72, which has been characterized as the most common mutation in the neurodegenerative disorders Amyotrophic Lateral Sclerosis (ALS) and Frontotemporal Dementia (FTD). While healthy individuals possess no more than 23 repeats, patient DNA samples were found to harbor hundreds of them [330,331]. Importantly, the G_4_C_2_ motif was shown to fold into a G-quadruplex structure from RNA molecules in vitro [332,333], suggesting a causal link between higher levels of this secondary structure in the transcript and pathology. Indeed, it was demonstrated that ALS-associated G4 levels in C9ORF72 mRNA drive the assembly of cytosolic membrane-less organelles, both in vitro and in cells. The over-representation of RNA granules was proposed to sequester key proteins and alter RNA metabolism, ultimately leading to neurodegeneration [267]. This hypothesis was demonstrated for hnRNP H, which was found to colocalize with G-quadruplex structures and accumulate in insoluble aggregates in C9-ALS patient cells and brain samples [263]. Importantly, this recruitment was correlated to an abnormal level of incorrectly spliced transcripts in ALS patient brain samples, supporting the idea that abnormally high levels of RNA granules associated with critical protein factors lead to aberrant splicing events in transcripts of genes previously implicated in ALS pathophysiology [263]. An additional hypothesis indicates that enzymatic degradation of long transcripts bearing expanded CGG repeats may not be sufficient to repress cytotoxicity. Indeed, it was recently demonstrated that even short G-rich RNAs drive liquid-to-solid phase transition through the self-assembly capabilities of G-quadruplex structures. The resulting RNA foci induced cell dysfunction and death, suggesting an alternative molecular pathway behind the pathophysiology of repeat expansion disorders [268]. Overall, expansion of G-quadruplexes was clearly designated as the causal factor in ALS and FTD using cellular models and patient-derived tissue samples. G4 motif repeat expansion in pathogenic alleles was shown to induce widespread deregulation of RNA metabolism, particularly in the context of splicing, through excessive RNA aggregate formation and sequestration of RNA-binding proteins. While alternative mechanisms linked to disease onset should not be overlooked, the current model reveals the central role of G-quadruplex structures in neurodegeneration, also pointing at G4s as amenable targets in ALS/FTD therapy.

The involvement of G-quadruplexes in human disease is especially relevant when considering these guanine assemblies as threats to genome integrity. In fact, as analyzed previously, defects in key components of the cellular response to G-quadruplex-induced polymerase stalling promote the formation of DNA lesions and increase chromosome fragility at pathological levels. It is no wonder that several conditions related to increased genome instability have been linked to impaired regulation of G-quadruplexes, as is the case of Bloom syndrome. Its symptoms include primordial dwarfism, sunlight sensitivity, impaired fertility, immunodeficiency, and a high incidence of cancers [221,334]. This rare genetic condition is caused by mutations in the *BLM* gene, which encodes a RecQ family helicase that is involved in multiple steps of DNA metabolism including G-quadruplex unwinding [221,223,224]. Importantly, a transcriptomics study of BLM-deficient fibroblasts revealed a positive correlation between upregulated genes and transcripts bearing putative G4 motifs (PQS), suggesting that BLM depletion results in a G-quadruplex-dependent global deregulation of transcription [230]. An alternative model of Bloom syndrome pathology proposes that the unusually high rates of recombination events and sister chromatid exchanges observed in BLM-deficient patients are caused by G4-mediated genomic instability, as has been shown for transcribed regions [235]. It is important to note that Bloom syndrome defects have not been exclusively linked to deregulation of G-quadruplexes. In fact, the BLM helicase was shown to contribute to several other G4-independent DNA-repair and recombination processes [335,336,337]. Despite this, it is undoubtedly possible that genomic loci prone to form G-quadruplexes are much harder to replicate in the absence of this critical helicase, thus ascribing part of the genome instability observed in patient cells to deregulated guanine tetraplexes. 

Most of the research into G-quadruplexes as drug targets, however, has been performed in the context of cancer. It has long been demonstrated that genomic instability is a hallmark of cancer, a desirable trait that dramatically increases genetic variability among the tumor population. This critical element enables the insurgence of mutations that are favorable to cancer cells and confer selective advantage, particularly in the six hallmarks of cancer postulated by Hanahan and Weinberg [338]: replicative immortality, resistance to cell death, self-sustaining proliferative signaling, evasion of growth suppressors, angiogenesis induction, active tissue invasion, and metastasis. At the heart of all this, poorly functional DNA damage response, cell cycle checkpoints, and/or apoptosis mechanisms are often associated with tumorigenesis. Remarkably, global G-quadruplex levels are generally higher in tumor or immortalized cells than in normal ones [85,339]. Although it is still unclear if guanine tetraplexes are themselves causal agents of cancer or simply a byproduct of neoplastic transformation, several lines of evidence point at these structures being amenable therapeutic targets. Firstly, as discussed above, specific conformations of folded G4s have been shown to inhibit telomerase [13], which is activated in cancer cells to maintain telomeres. Secondly, as explored previously, G-quadruplexes are endogenous factors that promote polymerase stalling and genomic instability, particularly in genetic backgrounds where G4 unwinding or bypassing has been compromised. Moreover, several promoters of oncogenes bear G4 motifs and folded G-quadruplex structures that have been linked to transcriptional regulation of these genes, including *c-myc*, *c-kit*, SRC, and KRAS [61,166,171,240]. It is no surprise that significant effort has been devoted to the discovery and design of small molecules capable of recognizing G-quadruplexes, as already discussed in a previous section. Indeed, a subset of these compounds successfully impaired telomerase activity, reduced oncogene expression, induced lethal DNA lesions and/or cell death, and counteracted cancer-specific phenotypes in various cell lines and a xenograft model [150,159,171,182] (reviewed in ref. [340]). Despite the growth in characterized G-quadruplex binders and the increased cytotoxicity of newer compounds, few have entered phase I clinical trials owing to their limited potency [341]. The candidate drug CX-5461 was recently shown to elicit a response in 14% of study participants carrying germline mutations of homologous recombination (HR) factors (BRCA1, BRCA2, or PALB2). Although limited, the data suggest that chemical G-quadruplex stabilization may have higher efficiency in patients with impaired HR [341,342]. On the whole, three main caveats might explain the poor performance of G-quadruplex binders in drug development. Firstly, from a pharmacological point of view, the decade-old drug design strategy that favors π-π stacking of ligands onto G-quartets fundamentally hinders the ability of the drug to be soluble in aqueous solutions, thus compromising the pharmacokinetics of most G-quadruplex ligands. Moreover, until recently, the limited availability of structural data and of specific assays made the design of molecules targeting a specific G-quadruplex within a genomic locus of interest (i.e., oncogene promoters) extremely difficult. In fact, it is not easy to demonstrate that transcriptional downregulation of a target gene is caused by stabilization of a precise G-quadruplex in its regulatory sequence rather than an indirect effect of global G4 persistence [248]. Luckily, the structural gap has partially been filled to allow the drug optimization process to be tailored to a specific G-quadruplex, provided it possesses unique topological characteristics [182]. On the other hand, non-selective G4 stabilization might still prove useful to globally deregulate transcription and induce DNA damage at such levels that induce cell death. In this case, however, it must be remembered that our understanding of G-quadruplex biology is still limited; therefore, predicting the outcome of guanine tetraplex stabilization is difficult and the process may produce unwanted consequences. Surely, expanding our knowledge on the topic will also help in defining new therapeutic strategies that target G-quadruplexes. A noteworthy example is the recently proposed link between chemical G-quadruplex stabilization and activation of the cGAS/STING pathway as a possible exploit to trigger innate immunity against tumors [341,343]. Overall, the progression of the G-quadruplex biology field is the foundation upon which new DNA- and RNA-targeting therapeutic strategies against cancer and genetic disorders may be discovered and optimized. Hence, once again the need to identify and characterize the factors and pathways that co-operate to modulate G-quadruplex levels in healthy cells is crucial in advancing the whole field.

## 7. Conclusions and Future Perspectives

Throughout seven decades of research, the relevance of the G-quadruplex in nucleic acid metabolism has been greatly expanded upon. Technological advancements, particularly next generation sequencing (G4-seq [121]), coupled to in-depth structural and dynamics investigations of these structures, both in vitro and in cells, have revealed crucial aspects of these polymorphic non-canonical secondary structures. G4s have been observed or predicted to exist in all currently sequenced genomes—from viruses to humans [110,111,122,190,199,203,205,208,209]. When combining all the experimental evidence, G-quadruplex biology has been implicated in all major processes of nucleic acid metabolism and to several layers of gene expression regulation, acting both as a positive and negative influences in any of those. Broadly speaking, G-quadruplexes influence cell biology in two major ways, one of which entails protein recruitment at specific genomic loci. The specificity of G-quadruplexes as scaffolding hubs for protein localization could, in theory, be guaranteed by the fact that G-quadruplexes are folded and unfolded in a controlled manner in cells. Additionally, multiple G4 binding proteins were shown to have higher binding affinity for a specific conformation of guanine tetraplexes, pointing at the relevance of the extensive structural polymorphism of these assemblies. However, it is currently unclear whether cells can dictate the topology of folding G-quadruplexes, for example, by locally regulating the pH or availability of stabilizing cations. Nevertheless, the spectrum of physiological functions regulated by G-quadruplexes is destined to expand as the G-quadruplex interactome does, again pointing at the central importance of discovering and characterizing new G4 binding factors in this research field. On the other hand, G-quadruplexes were shown to work independently of protein recruitment, in particular as obstacles to polymerase activity. In this sense, G-quadruplexes become endogenous determinants of DNA lesions and genomic instability. In this case, the molecular mechanisms behind the prevention, formation, and resolution of G4-dependent DNA damage are yet to be elucidated completely. In the future, the discovery of factors that recognize G-quadruplexes threatening genome integrity and/or guide DNA-damage-processing factors into a specific repair pathway will be paramount in dissecting the role of guanine tetraplexes in physiology and disease.

While the current comprehensive view of G-quadruplex biology requires key information on molecular mechanisms, therapeutic strategies either targeting or employing guanine tetraplexes have been explored since the 1990s. Their inhibitory effect on either oncogene expression or directly on telomerase activity has propelled anticancer research into designing increasingly more specific and cytotoxic G4 ligands, with limited clinical applicability so far. With time, however, alternative strategies have been proposed, from targeting G4 binding proteins as a drug delivery system [344] to stabilizing guanine tetraplexes for the stimulation of innate immunity mechanisms [341,343]. An intriguing avenue yet to be fully explored is G4 unwinding by way of destabilizing ligands [185], whose applicability in cases of aberrantly higher tetraplex levels—especially in genetic conditions such as ALS, FTD, or Bloom syndrome—is predicted to be beneficial. On the whole, the potential translational applications of G-quadruplex studies are intriguing. The limited success of recent clinical trials of G4 ligands underlines the crucial need to dissect G4 biology at the molecular level in order to identify clinically relevant cellular pathways to be more efficiently targeted in affected patients. In this context, expanding the list of factors that specifically interact with guanine tetraplexes will provide new insight into how these peculiar secondary structures can influence the wide variety of biological pathways that have been presented in this review. An additional avenue for future research involves the precise deciphering of the relevance of G-quadruplex polymorphism in cells, which could be theoretically achieved thanks to the expanding list of G4 ligands capable of recognizing a specific tetraplex topology or of inducing a switch between these conformations [184].

## Figures and Tables

**Figure 1 ijms-25-03162-f001:**
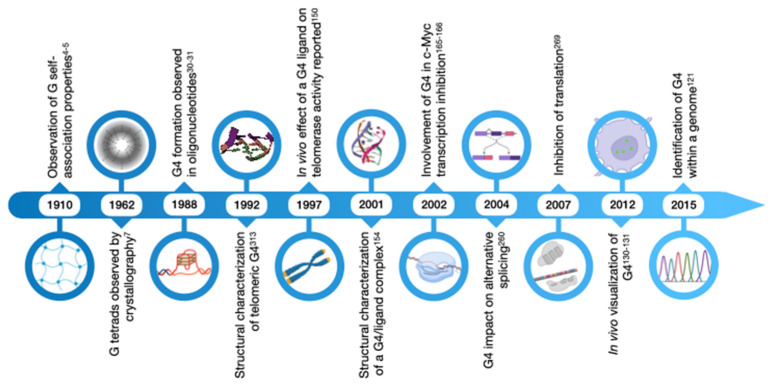
Selected milestones in G-quadruplex research.

**Figure 3 ijms-25-03162-f003:**
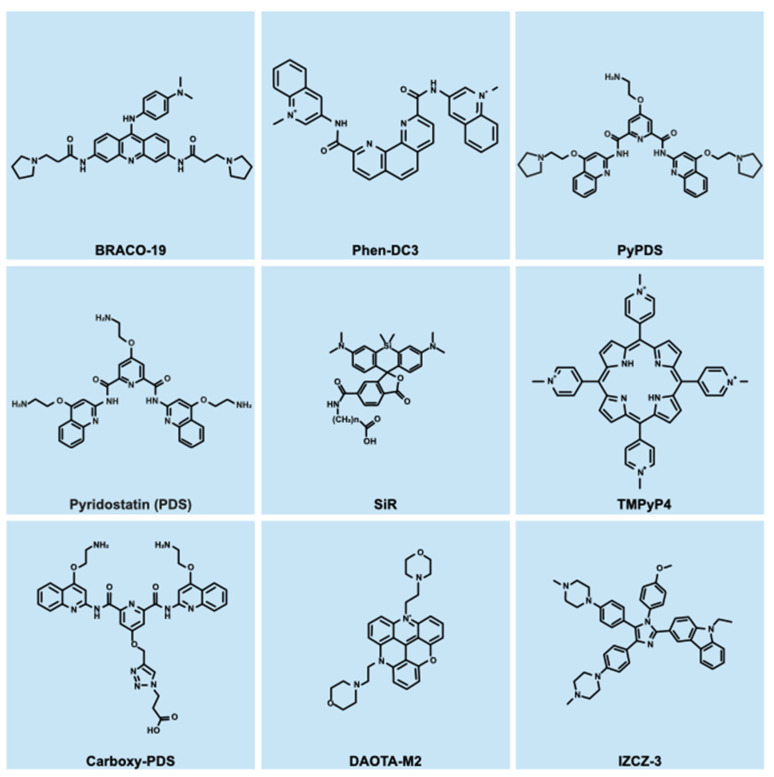
Chemical structures of G-quadruplex ligands mentioned in the text, as retrieved from PubChem [173], such as Braco-19 (CID: 9808666; https://pubchem.ncbi.nlm.nih.gov/compound/9808666), Phen-DC3 (CID: 44449504; https://pubchem.ncbi.nlm.nih.gov/compound/44449504), PDS (CID: 25227847; https://pubchem.ncbi.nlm.nih.gov/compound/25227847), TMPyP4 (CID: 135442972; https://pubchem.ncbi.nlm.nih.gov/compound/135442972) and IZCZ-3 (CID: 137628645; https://pubchem.ncbi.nlm.nih.gov/compound/137628645); PyPDS and SiR have been successfully fused into SiR-PyPDS [174]. Carboxy-PDS, while structures of Carboxy-PDS and DAOTA-M2 were taken from their respective papers [131,175].

**Figure 4 ijms-25-03162-f004:**
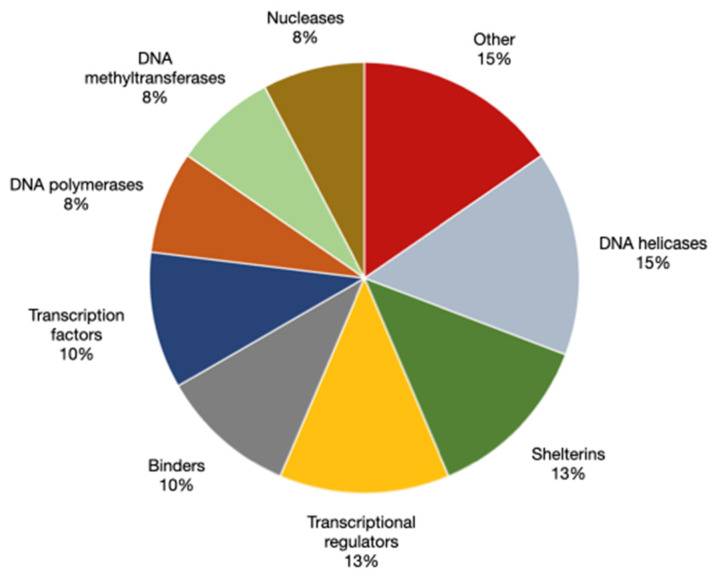
Classes of DNA G-quadruplex binding proteins listed in G4IPD.

**Figure 5 ijms-25-03162-f005:**
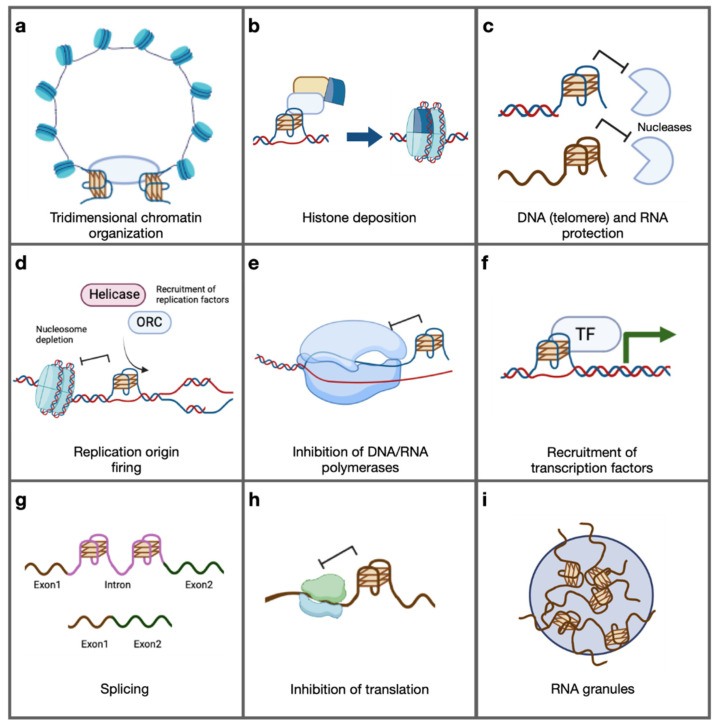
Cellular roles of G-quadruplexes. Thanks to their capability to distort the nucleic acid structure and to interact with a plethora of proteins, G-quadruplexes have been ascribed a role in several key cellular processes, comprising, but not limited to organization of chromatin in topologically associating domains (TADs) (**a**), deposition of non-replicative histone variants (**b**), inhibition of DNA and RNA nucleases (**c**), firing of replication origin (**d**), inhibition of replication and transcription (**e**), regulation of transcription (**f**), RNA maturation (**g**), regulation of translation (**h**), and formation of RNA stress granules (**i**).

**Table 1 ijms-25-03162-t001:** List of in vitro, in silico, and in vivo techniques commonly used to study G-quadruplexes.

Technique	Data Obtained	Type	Advantages	Disadvantages
X-ray crystallography	Structure Ligands	in vitro	Angstrom level resolution	Requires suitable G-quadruplex crystals Impossible to study dynamics
NMR spectroscopy	Structure Stability (time) Ligands	in vitro in vivo (adapted)	Physiological-closed conditions Dynamic studies Can detect multiple G4 at the same time	Limited sensitivity
CD spectroscopy	Structure Stability (temperature)	in vitro	Can discriminate between parallel/antiparallel/hybrids G4s Cost-effective study of ligand stabilization/destabilization	Less informative than NMR or X-ray Susceptible to non-canonical conformations Susceptible to the presence of A-form duplexes
UV melting	Stability (temperature) Ligands	in vitro	Dynamic studies	Low resolution
FRET	Stability Distance between 3′ and 5′ ends of ssDNA	in vitro	Absolute distance measurement Single molecule resolution	Fluorophores might affect G4 folding
Bioinformatic prediction	Prediction of G4 within the genome/transcriptome	in silico	Cost-effective Genome-wide analyses	Can only predict G4s matching the model used to generate the predictor Requires validation
G4-seq and rG4-seq	Distribution in genome and transcriptome	in vivo	Can identify non-canonical G4s	Require validation Susceptible on the type of the used molecule
Antibody-based methods	G4 spatial distribution ChIP	in vivo	Direct visualization of G4s in cells	Susceptible on the type of the used molecule

## Data Availability

Not applicable.

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
