# Peer review of "Spotlight on G-Quadruplexes: From Structure and Modulation to Physiological and Pathological Roles"

_ijms, 2024, doi:10.3390/ijms25063162_

Round 1

Reviewer 1 Report

Comments and Suggestions for Authors

The review manuscript from Maria Chiara Dell’Oca et al., "Spotlight on G-quadruplexes: from structure and modulation to physiological and pathological roles", gives a decent overview of the many years of research on various aspects of G-quadruplexes ranging from their basic (and diverse) organization to an extensive description of their biological roles, regulation, pharmaceutical aspects, mechanisms and interactions. In general the review provides a nice and plain summary on these fascinating and exceptional biological structures. Only a few minor points could be taken into account.

Even though the authors mention the relevance of polymorphism and the biological impact of G-quadruplexes in the cell is described in very much detail, fundamental structural studies are only briefly discusses and unfortunately most recent structural studies on the (un)folding dynamics and kinetics seem to be missing (most references in this section date back from more than 10 years ago).

Furthermore, it seems that the terms "G-quadruplex", "tetraplex" and "G4" are throughout the review used alternatingly to describe the same thing. If that is not the case please describe these terms and their differences.

With an little update on the recent structural techniques and discoveries, I would propose that the manuscript is acceptable for publication.

Author Response

The review manuscript from Maria Chiara Dell’Oca et al., "Spotlight on G-quadruplexes: from structure and modulation to physiological and pathological roles", gives a decent overview of the many years of research on various aspects of G-quadruplexes ranging from their basic (and diverse) organization to an extensive description of their biological roles, regulation, pharmaceutical aspects, mechanisms and interactions. In general the review provides a nice and plain summary on these fascinating and exceptional biological structures. Only a few minor points could be taken into account.

Even though the authors mention the relevance of polymorphism and the biological impact of G-quadruplexes in the cell is described in very much detail, fundamental structural studies are only briefly discusses and unfortunately most recent structural studies on the (un)folding dynamics and kinetics seem to be missing (most references in this section date back from more than 10 years ago).

We thank the reviewer for reading and commenting on our manuscript.

Following his suggestions, we have added more recent techniques at the end of section 3.1, highlighted in red, with updated references.

Furthermore, it seems that the terms "G-quadruplex", "tetraplex" and "G4" are throughout the review used alternatingly to describe the same thing. If that is not the case please describe these terms and their differences.

We have changed the following sentence on line 63 to make it clearer that we are using these terms as synonyms

Nowadays, the consensus for intramolecular G-quadruplex (also known as G4 or tetraplex)

Reviewer 2 Report

Comments and Suggestions for Authors

The review authored by Maria Chiara Dell’Oca et al. is a very nice contribution to the field of G-quadruplexes (G4s). It provides a comprehensive synthesis of knowledge in diverse areas of G4s research. In the introductory chapter, historical milestones in G4 research are presented, then the G4 structure (and various G4 types/conformations) is described in detail. The following chapter is focused on available tools for studying G4s, comprising both in vitro, in vivo, and bioinformatic approaches. Then, the prevalence of G4s across available genomes of various organisms (including viruses) is discussed. The next chapters shed light on the regulatory mechanisms of G4s in cells and also link G4s to physiological processes as well as severe human diseases. The last chapter provides some future perspectives and conclusions. Overall, the review is very well conceptualised and will be useful for the G4 community.

Specific points to be improved:

Line 11 ... I would replace "non-B secondary structures" with "noncanonical structures", as RNA is non-B by default (and G4s are presented both in DNA and RNA, as you discuss below)

Line 41 ... "as two scientists" ... which ones? references 4-6 are published by 5 authors, so it should be specified or edited

Chapter 2 ... also handedness of G4s should be explicitly mentioned, i.e. the existence of right and left-handed G4s ... reference 34 (already cited). In addition, in the whole manuscript, I haven't found any mention of i-motifs - usually, it is believed that i-motifs can arise on opposite C-rich strands of G4s. This issue should be mentioned as well and some reference(s) added.

Figure 2a ... the G4 pattern should be expressed much more precisely ... why G3-G5? In study https://pubmed.ncbi.nlm.nih.gov/29733879/ (should be cited), it was found that sequences for six-stacked G-quadruplexes are present in the human genome as well (and within very interesting genes). Also, it cannot be excluded that 7+ stacked intramolecular G4s exist, either in human or in other organisms. The same with two stacked G4s. Also loop length formulation is too strict, why limited to the length of 7 nucleotides? I would try to present some more universal formula for the G4-forming sequence.

Line 135 ... "NMR structures of G-quadruplexes have been found in human minisatellites CEB1 and CEB25" ... should be reformulated, something like "G4s structures were *solved* by NMR"

Table 1 ... I am missing some structural approaches in silico, like G4 structure prediction via e.g. 3DNus web server and also docking and molecular dynamics (e.g. for inspecting G4 interaction with protein(s))

Around line 290 ... except for pattern searches and neural networks, also algorithm based on G richness and skewness (G4Hunter) should be mentioned

Line 301 ... on the contrary, is there also information about how many predicted PQSs were not caught by G4-seq? Should be mentioned here. Also, consider false negative and false positive rates of G4-seq ... we largely know very little about it 

Line 551-552 ... should be formulated better, as CDS is a subset of mRNA (without introns, UTRs)

Line 564 ... HCMV abbreviation should be explained

Lines 604 - 615 ... what about phylogenetic conservation of particular G4 loci? See e.g. https://www.mdpi.com/1422-0067/22/14/7381. Are G4s rather conserved or dynamic between closely/distant related species? I mean particular loci, not overall frequencies of G4-forming sequences in genomes

Chapter 5.2 ... maybe some scheme would be beneficial to visually depict classes of G4 binding proteins (i.e. stabilizing, destabilizing, helicases, readers, ...)

Line 772 ... TADs abbreviation needs to be explained here (first mention)

Line 988 ... t-loop should be T-loop (previously you mentioned G-loops and R-loops)

Line 1032 ... Frontal-Temporal Dementia should be Frontotemporal Dementia or Fronto-Temporal Dementia, Frontal sounds strange to me

Line 1068 ... BLM gene, genes should be in italics

References should be double-checked and names of journals unified, either abbreviations or full names

Author Response

Reviewer #2:

The review authored by Maria Chiara Dell’Oca et al. is a very nice contribution to the field of G-quadruplexes (G4s). It provides a comprehensive synthesis of knowledge in diverse areas of G4s research. In the introductory chapter, historical milestones in G4 research are presented, then the G4 structure (and various G4 types/conformations) is described in detail. The following chapter is focused on available tools for studying G4s, comprising both in vitro, in vivo, and bioinformatic approaches. Then, the prevalence of G4s across available genomes of various organisms (including viruses) is discussed. The next chapters shed light on the regulatory mechanisms of G4s in cells and also link G4s to physiological processes as well as severe human diseases. The last chapter provides some future perspectives and conclusions. Overall, the review is very well conceptualised and will be useful for the G4 community.

 We truly thank the reviewer for his/her appreciation of the work, and for the relevant suggestions that help us to improve the work.

Specific points to be improved:

Line 11 ... I would replace "non-B secondary structures" with "noncanonical structures", as RNA is non-B by default (and G4s are presented both in DNA and RNA, as you discuss below)

We thank the reviewer for pointing this out, we've fixed it.

Line 41 ... "as two scientists" ... which ones? references 4-6 are published by 5 authors, so it should be specified or edited

We have edited the text as: when was reported

Chapter 2 ... also handedness of G4s should be explicitly mentioned, i.e. the existence of right and left-handed G4s ... reference 34 (already cited).

We thank the reviewer for pointing this out, we've added this sentence in the text in section 2.1 line 101:

[32]). While most G-quadruplexes have been reported to have a right-handed helical twist, left-handed G4s[33] and even hybrid G4s[34] have also been observed. Left-handed G-quadruplexes can be formed by at least two distinct minimal sequence motives of 12 nt[35,36]. Interestingly, the GTGGTGGTGGTG motif, that is highly abundant in the human genome, was not only shown to independently form left-handed G4 structures, but also to drive the formation of left-handed conformations from several other sequences, when attached to them[35]. Machine-learning methods to classify right- and left-handed G4s based on torsional angles have been recently explored[37].

In addition, in the whole manuscript, I haven't found any mention of i-motifs - usually, it is believed that i-motifs can arise on opposite C-rich strands of G4s. This issue should be mentioned as well and some reference(s) added.

We thank the referee for this insight. We have added in section 2.1 line 163 this part:

However, the opposite is true when considering the case of i-motifs. Similarly to guanine stratches, a series of cytidine residues in DNA or RNA can also associate in four-stranded structures called i-motifs. These are secondary nucleic acid structures consisting of four strands stabilized by hemi-protonated and intercalated cytosine base pairs (C:C+)[75,76]. It has recently been widely demonstrated that the complementary strand of any G-quadruplex forming sequence is prone to forming i-motifs (reviewed in ref. [75,77]). Importantly, the stabilization of G-quadruplexes using small molecules was shown to destabilize the i-motifs, and vice versa. This suggests that these structures are interdependent[78]. Owing to their distinctive physicochemical properties, these i-motif structures have garnered considerable attention as novel targets for drug development (reviewed in ref. [79]), however a thorough discussion of these secondary structures is beyond the scope of this review.

Figure 2a ... the G4 pattern should be expressed much more precisely ... why G3-G5? In study https://pubmed.ncbi.nlm.nih.gov/29733879/ (should be cited), it was found that sequences for six-stacked G-quadruplexes are present in the human genome as well (and within very interesting genes). Also, it cannot be excluded that 7+ stacked intramolecular G4s exist, either in human or in other organisms. The same with two stacked G4s. Also loop length formulation is too strict, why limited to the length of 7 nucleotides? I would try to present some more universal formula for the G4-forming sequence.

We thank the referee for this point. We have modified the Figure 2a using a more general G≥3NxG≥3NxG≥3NxG≥3

Line 135 ... "NMR structures of G-quadruplexes have been found in human minisatellites CEB1 and CEB25" ... should be reformulated, something like "G4s structures were *solved* by NMR"

 We've fixed it.

Table 1 ... I am missing some structural approaches in silico, like G4 structure prediction via e.g. 3DNus web server and also docking and molecular dynamics (e.g. for inspecting G4 interaction with protein(s))

Thank you for suggesting this, we have added the following sentence to the text:

Additionally, other bioinformatic tools have been used to predict the three-dimensional conformation of G-quadruplexes from a given sequence. Indeed, inter- and intramolecular G-quadruplex structures can be generated in silico using web tools such as 3D-NuS[118]. This modelling could be used to explore the dynamics of different G-quadruplexes and the docking of proteins and ligands.

Around line 290 ... except for pattern searches and neural networks, also algorithm based on G richness and skewness (G4Hunter) should be mentioned

On line 292 this phrase has been added:

Other detection tools as G4Hunter assign the score using G richness and G skewness[115]               

Line 301 ... on the contrary, is there also information about how many predicted PQSs were not caught by G4-seq? Should be mentioned here. Also, consider false negative and false positive rates of G4-seq ... we largely know very little about it 

We thank the referee for this point. We have modified the text adding this interesting analysis:

Furthermore, the efficacy of some PQS prediction algorithms was tested using the output of the high-throughput G4-seq method[119], which consists of a modified Illumina sequencing protocol where polymerase arrest upon encountering a folded G-quadruplex in the template is identified by a drop in sequencing quality scores. Only the dip in Q-scores specific for G-quadruplexes was selected thanks to the comparison between a sequencing run in standard conditions and one where G-quadruplex stabilizing factors (either K+ or the small molecule ligands as pyridostatin or PhenDC3 were added to the sequencing buffer. G4-seq of the human genome identified more than 700000 G-quadruplex forming sites, of which about 70% were not predicted by the standard G4 motif-based Quadparser algorithm[108,119], possibly comprising noncanonical G4 structures escaping the algorithm folding rule (G3+N1–7G3+N1–7G3+N1–7G3+), like those with loops longer than 7 bases, bulges in the G-tracts or with only two G-tetrads. On the contrary, the original G4-seq method failed to detect 27% in PDS and 40% in K+ of Quadparser predicted canonical G4-forming motifs[119]. Even when considering predicted potential G4-forming motifs with loop lengths up to 12 nucleotides, 37% of them still evaded G4-seq in PDS outputs[120]. This inconsistency may be mainly explained by inadequate sequencing coverage in certain GC-rich genomic regions (due to inefficient amplification at stable G4s in Na+ and PCR biases during library preparation), PDS stabilization performed in Na+, and low resolution of the observed G4 motifs and consequent merge of proximal G4 motifs of the original experiment. Additionally, limited G4 stability, binding specificities of the employed G4 ligands and the in vitro experimental conditions may account for a fraction of false negatives. To overcome these limitations, improvements have been introduced in the second-generation method that have successfully increased the specificity of the assay in K+ PDS. In fact, the refined protocol detected ∼95% of human canonical Quadparser G4s and 84% of the ~706 k potential G4-forming sequences with loops as long as 12 nt[120].

Line 551-552 ... should be formulated better, as CDS is a subset of mRNA (without introns, UTRs)

We have modified the text has suggested:

The rG4-seq dataset on human transcriptome revealed that G-quadruplexes in mRNA have the highest density in 5’ and 3’ UTRs compared to CDS (in accordance with G4-seq data)[121]

Line 564 ... HCMV abbreviation should be explained

We've fixed it.

Lines 604 - 615 ... what about phylogenetic conservation of particular G4 loci? See e.g. https://www.mdpi.com/1422-0067/22/14/7381. Are G4s rather conserved or dynamic between closely/distant related species? I mean particular loci, not overall frequencies of G4-forming sequences in genomes

 Thank you for suggesting this, we have added the following sentence to the text:

Line 665

In literature, various cases of specific G4 loci that are phylogenetically conserved have been reported. To report some examples, a validated G4 locus in the RPB1 gene coding for the large subunit of RNA polymerase II has been found to be conserved in the Archaeplastida plant supergroup[208]. Additionally, a G-quadruplex forming sequence was found in the 5’UTR of the NRAS proto-oncogene’s mRNA in at least 6 different mammalian species, where it may have a role in translational regulation[209].

Chapter 5.2 ... maybe some scheme would be beneficial to visually depict classes of G4 binding proteins (i.e. stabilizing, destabilizing, helicases, readers, ...)

We have added a new Figure 4 for this.

Line 772 ... TADs abbreviation needs to be explained here (first mention)

We've fixed it.

Line 988 ... t-loop should be T-loop (previously you mentioned G-loops and R-loops)

We've fixed it.

Line 1032 ... Frontal-Temporal Dementia should be Frontotemporal Dementia or Fronto-Temporal Dementia, Frontal sounds strange to me

We've fixed it.

Line 1068 ... BLM gene, genes should be in italics

We've fixed it.

References should be double-checked and names of journals unified, either abbreviations or full names

We've fixed it.